# Pareto Optimal Risk Measure Agnostic Distributional Bandits with Heavy-Tail Rewards

**Kyungjae Lee**[1][*]   **Dohyeong Kim**[2]   **Taehyun Cho**[2]   **Chaeyeon Kim**[1]   **Yunkyung Ko**[1]
**Seungyub Han**[2]   **Seokhun Ju**[2]   **Dohyeok Lee**[2]   **Sungbin Lim**[1,3][†]

[1]Department of Statistics, Korea University
[2]Department of Electrical and Computer Engineering, Seoul National University
[3]LG AI Research

## Abstract

This paper addresses the problem of multi-risk measure agnostic multi-armed bandits in heavy-tailed reward settings. We propose a framework that leverages novel deviation inequalities for the 1-Wasserstein distance to construct confidence intervals for Lipschitz risk measures. The distributional LCB (DistLCB) algorithm is introduced, which achieves asymptotic optimality by deriving the first lower bounds for risk measure aware bandits with explicit sub-optimality gap dependencies. The DistLCB is further extended to multi-risk objectives, which enables Pareto-optimal solutions that consider multiple aspects of reward distributions. Additionally, we provide a regret analysis that includes both gap-dependent and gap-independent bounds for multi-risk settings. Experiments validate the effectiveness of the proposed methods in synthetic and real-world applications.

## 1 Introduction

The multi-armed bandit (MAB) framework provides a fundamental model for sequential decision-making under uncertainty [1]. Traditional approaches in the MAB aim to maximize expected rewards by balancing exploration and exploitation. These methods have found widespread applications in areas such as clinical trials or recommendation systems. However, they typically rely on the expected value as the sole performance criterion, which limits their ability to capture other important properties of reward distributions. This limitation becomes critical in risk-sensitive applications where rare but high-impact outcomes play a central role, such as financial crashes or adverse medical events.

Risk measure aware bandit frameworks aim to address this drawback by incorporating alternative metrics such as conditional value-at-risk (CVaR) [2–5], mean-variance [6, 7], spectral risk measures (SRM) [8, 9], and distortion risk measures (DRM) [10]. These methods allow decision-makers to prioritize different parts of the reward distribution, such as the lower tail, and are better suited to applications requiring safety, robustness, or downside control. However, most existing methods focus on optimizing a single risk measure, which still leaves important aspects of uncertainty unaddressed.

Many practical problems require decision-makers to account for multiple risk perspectives at the same time. For example, a portfolio manager must consider both expected returns and downside risks. In clinical decision-making, treatment effectiveness must be weighed against the risk of side effects. A toy example in Figure 1a and Table 1b illustrates how different risk measures prioritize distinct distributional properties. In such cases, identifying a single best action may not be appropriate. Instead, one must identify a Pareto-optimal set of actions that are not dominated under any of the chosen risk measures.

---

[*]Contact: {kyungjae_lee, sungbin}@korea.ac.kr
[†]Corresponding author.

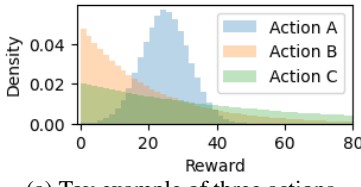

(a) Toy example of three actions.

| Action | Mean | $\text{CVaR}_{0.5}$ | $\text{CVaR}_{0.8}$ | Wang | P. Opt. |
|--------|------|---------|---------|------|---------|
| A | 25.0 | 19.4 | 15.2 | 21.0 | ✓ |
| B | 20.0 | 6.1 | 2.2 | 10.0 | |
| C | 50.0 | 15.1 | 5.3 | 24.7 | ✓ |
| Opt. Act. | C | A | A | C | |

(b) Summary of risk values.

Figure 1: (a) Reward histograms for three actions. (b) Summary of associated risk values. $\text{CVaR}_{0.5}$ and $\text{CVaR}_{0.8}$ are Conditional Value-at-Risk measures at the $50\%$ and $80\%$ quantiles, respectively. Wang [11] represents a distortion risk measure with $g(p) = p^2$. Opt. Act. shows the optimal action for each metric. P. Opt. marks actions that are Pareto-optimal. Action B is strictly dominated under all risk measures and thus not Pareto-optimal, while Actions A and C are each optimal under different risk criteria, making both Pareto-optimal in the multi-risk sense.

A growing body of work has begun to explore the estimation of multiple risk measures simultaneously. Notably, methods have been proposed for constructing confidence intervals for risk measure classes such as CVaR and distortion-based risks under bounded or sub-Gaussian noise [8–10]. For example, Zhang and Ong [12] considered quantile estimation in sub-Gaussian settings, while Cassel et al. [8], Tan et al. [9], Liang and Luo [10] derived confidence bounds for the Lipschitz risk class under light-tailed assumptions. These contributions lay groundwork for multi-risk optimization, but their applicability remains limited in heavy-tailed environments, which are commonly encountered in finance, healthcare, and other real-world domains [13–20].

To address this gap, we propose a unified multi-risk bandit framework that remains effective under heavy-tailed reward distributions. While prior works have studied multi-risk optimization under light-tailed assumptions, to the best of our knowledge, this is the first framework that supports general Lipschitz risk measures in multi-risk bandit problems under heavy-tailed noise. At the core of our approach is the median of empirical quantiles (MoEQ) estimator, which extends median-of-means techniques to quantile estimation. MoEQ enables robust estimation of distributional properties and supports a wide range of Lipschitz risk measures without requiring manual construction of risk-specific confidence intervals. Our contribution is summarized as follows:

- We propose a median of quantile estimation under heavy-tailed noise and introduce bootstrap resampling to mitigate its computational inefficiencies.

- We derive new deviation inequalities for the 1-Wasserstein distance, enabling the construction of confidence intervals for Lipschitz risk measures in heavy-tailed reward settings. This result extends recent works [9, 10].

- We introduce the distributional LCB (DistLCB) algorithm, which leverages MoEQ to provide regret guarantees across all Lipschitz risk measures. We further extend this to a multi-risk variant and present the first regret bounds for multi-risk bandits under heavy-tailed rewards.

- We also establish the explicit dependency of sub-optimality gaps in the asymptotic lower bounds for risk measure aware bandits with SRM and DRM. We derive both gap-dependent and gap-independent regret bounds for single and multi-risk objectives, confirming the optimality of our algorithms.

## 2   Related Work

risk measure aware bandit algorithms have been studied under various risk measures and distributional assumptions. Early works primarily focused on specific risk criteria such as CVaR or mean-variance under light-tailed rewards, yielding asymptotic or problem-dependent regret bounds [2, 3, 7, 21–23]. To generalize beyond single risk types, bandit algorithms for Lipschitz risk measures, including CVaR, spectral, and distortion-based risks, have been proposed, but remain limited to sub-Gaussian or bounded settings [8–10]. In contrast, works addressing heavy-tailed rewards largely focus on single-risk setting, such as the mean or CVaR [24–29, 4, 5], with only Bhatt et al. [30] analyzing the broader Lipschitz risk class under infinite variance regimes. Multi-risk formulations, which aim to optimize multiple risk measures simultaneously, have been explored in light-tailed settings [31–34],

| References | Multi-Risk | Risk Type | Distribution | Lower Bound |
|---|---|---|---|---|
| [2, 3, 7, 21–23] | ✗ | CVaR | Light Tails | Asymptotic [22] |
| [8–10] | ✗ | Lipschitz Risk Class | Light Tails | - |
| [24–29] | ✗ | Mean | Heavy Tails | - |
| [4, 5] | ✗ | CVaR | Heavy Tails | - |
| [30] | ✗ | Lipschitz Risk Class | Heavy Tails | Minimax |
| [31–34] | ✓ | Mean | Light Tails | - |
| Ours | ✓ | Lipschitz Risk Class | Heavy-tails | Asymptotic, Minimax |

Table 1: Comparison of representative risk measure aware bandit algorithms. The Lipschitz risk class includes mean and CVaR. Heavy-tailed distributions include light-tailed ones. Multi-Risk indicates whether multiple risks are jointly optimized; Lower Bound indicates whether regret lower bounds are analyzed.

but remain undeveloped for general Lipschitz risks or heavy-tailed environments. We address this gap with a unified framework for multi-risk optimization under heavy-tailed rewards, enabled by the MoEQ estimator with exponential convergence guarantees. This landscape is summarized in Table 1, which categorizes existing methods by their support for multi-risk objectives, risk types, distributional assumptions, and lower bound analyses. A more detailed comparison with prior work is provided in Appendix B.

## 3 Preliminaries

In this section, we introduce the foundational concepts of risk measure aware bandit problems with heavy-tailed rewards, Lipschitz risk measures, and Wasserstein distance.

### 3.1 risk measure aware Bandits with Heavy-Tailed Rewards

Consider a set of $K$ actions, $\mathcal{A} := \{1, 2, \cdots, K\}$, each associated with an unknown reward distributions, $\{\mu_a\}_{a \in \mathcal{A}}$. At each time $t$, the learner chooses an action $A_t$ by following some policy $\pi_t$ and observes a random reward $X_{A_t} \sim \mu_{A_t}$ for the selected action. In risk measure aware bandits, the objective is to minimize a risk measure aware criterion rather than focusing solely on expected rewards. A risk measure $\rho(X_a)$ evaluates the *risk* of each action and captures aspects of the reward distribution beyond the mean, such as variability or tail behavior. The cumulative regret of risk measure aware bandits [9, 30] is defined as

$$\mathcal{R}_n^\rho := \mathbb{E}\left[\sum_{t=1}^n \rho(X_{A_t}) - \rho(X_{a_*})\right], \tag{1}$$

where $a_*$ denotes the optimal action, i.e., $\rho(X_{a_*}) = \min_{a \in \mathcal{A}} \rho(X_a)$. While traditional bandit problems *maximize* rewards, this paper sets the objective to *minimize* cumulative risk for risk-sensitive decision-making. We also consider the heavy-tail assumption on the reward distributions:

**Assumption 3.1** (Heavy-Tailed Distribution and Bounded $p$-th Moment). For each action $a \in \mathcal{A}$, assume that $\mu_a$ belongs to the heavy-tailed class $\mathcal{H}(p)$ for some $p \in (1, 2]$, defined as: $\mathcal{H}(p) := \{\mu : \mathbb{E}_{X \sim \mu}[|X|^p] \leq \nu_p\}$, where $\nu_p$ is a finite constant.

This assumption reflects the presence of heavy-tailed distributions, such as Pareto, Fréchet, or Weibull, where moments higher than $p$ does not exist, necessitating robust estimation methods [24, 26].

### 3.2 Lipschitz Risk Meausre

Risk measures quantify various aspects of uncertainty in reward distributions. Denote the space of all random variables with a finite expectation as $\mathcal{L}_1$. A risk measure $\rho$ maps a random variable in $\mathcal{L}_1$ to a real number. To handle multi-risk settings, it is necessary to estimate a class of risk measures rather than a single measure. We consider Lipschitz continuous risk measure with respect to distance metric on $\mathcal{L}_1$, which provides a broad framework for multiple risks, as in Tan et al. [9].

**Definition 3.2** ($\kappa$ Lipschitz Risk Measure w.r.t. Distance $\mathbb{D}$). A risk measure $\rho$ defined on random variables is said to be $\kappa$ Lipschitz continuous with respect to a distance $\mathbb{D}$ on $\mathcal{L}_1$ if there exists a constant $\kappa > 0$ s.t. for all $X, Y \in \mathcal{L}_1$, $|\rho(X) - \rho(Y)| \leq \kappa \mathbb{D}(\mu_X, \mu_Y)$.

Lipschitz risk measure is a general class of risk meausures including CVaR, spectral risk measures (SRM), distortion risk measures (DRM), utility-based shortfall risk (UBSR), or certainty equivalent risk measure (CERM). See Appendix C for more details.

In analyzing Lipschitz risk measures, the choice of distance metric plays a crucial role. Commonly used metrics such as $\ell_1, \ell_2, \ell_\infty$, and the 1-Wasserstein distance have been primarily analyzed under bounded support, sub-Gaussian, and sub-Exponential assumptions [21, 9, 10, 30]. In this paper, we focus on the 1-Wasserstein distance, which has been less explored in heavy-tailed settings.

**Definition 3.3** (1-Wasserstein Distance). Let $F$ and $G$ be cumulative distribution functions (CDFs) supported on $\mathbb{R}$, and let $F^{-1}$ and $G^{-1}$ denote their corresponding quantile functions (i.e., inverse CDFs). The 1-Wasserstein distance between $F$ and $G$ is defined by

$$W_1(F, G) := \int_{-\infty}^{\infty} |F(x) - G(x)| \, dx = \int_0^1 |F^{-1}(t) - G^{-1}(t)| \, dt. \tag{2}$$

### 3.3 Multi-Risk Bandit Problems

In classical risk measure aware bandit problems, a single risk measure evaluates actions but offers a limited perspective on the reward distribution. Multi-risk bandit problems address this by considering $r$ number of Lipschitz risk measures, $\boldsymbol{\varrho} = (\rho_1, \rho_2, \cdots, \rho_r)$, with their corresponding Lipschitz constants $\mathcal{K} := (\kappa_1, \kappa_2, \cdots, \kappa_r)$ to capture diverse risk characteristics such as variability and tail behavior. The objective is to identify the Pareto optimal set of actions, $\mathcal{P}_*$, where no action dominates another across all risks [31, 32, 34]. The dominance relationship is defined as follows:

**Definition 3.4** (Dominance). Let $u$ and $v$ be a vector in $\mathbb{R}^r$.

- $u$ dominates $v$, $u \succ v$: For all $d \in [r]$, $u_d \geq v_d$, and there exists at least one $d'$ s. t. $u_{d'} > v_{d'}$.

- $u$ is non-dominated by $v$, $u \nprec v$: There exists a dimension $d \in [r]$ such that $u_d > v_d$.

- $u$ is incomparable with $v$, $u \| v$: Neither $u \succ v$ nor $v \succ u$.

Formally, the Pareto optimal set is defined as $\mathcal{P}_* := \{a \in \mathcal{A} \mid \forall a' \in \mathcal{A} \setminus \{a\}, \boldsymbol{\varrho}(X_{a'}) \nprec \boldsymbol{\varrho}(X_a)\}$. To analyze the convergence rate to the Pareto optimal set, we barrow the mathematical framework in Drugan and Nowe [31]. The sub-optimality gap of arm $a$ is defined as

$$\Delta_a^{\text{Pareto}} := \inf_{\epsilon \geq 0} \{ \epsilon \mid \boldsymbol{\varrho}(X_a) + \epsilon \mathbf{1} \| \boldsymbol{\varrho}(X_{a'}), \forall a' \in \mathcal{P}_* \}, \tag{3}$$

where $\mathbf{1}$ denotes a vector of $r$ ones. Note that if $a \in \mathcal{P}_*$, then, $\Delta_a^{\text{Pareto}} = 0$ holds. Then, the regret can be defined as

$$\mathcal{R}_n^{\text{Pareto}} := \sum_{a \in \mathcal{A}} \Delta_a^{\text{Pareto}} \mathbb{E}\left[T_a(n)\right]. \tag{4}$$

Multi-risk bandits are especially important in heavy-tailed settings, where multiple risk measures can highlight wide spectra of events more effectively.

## 4 Median of Empirical Quantiles for Heavy-Tailed Distribution

We introduce the median of empirical quantiles (MoEQ), a robust method for estimating reward distributions in heavy-tailed settings. From the definition of Lipschitz risk measures, the confidence interval depends on the distance between the empirical and true distributions. Constructing reliable confidence intervals under heavy-tailed distributions requires robust quantile estimation. MoEQ addresses this by ensuring stable estimates and allows accurate computation of various risk measures within the Lipschitz risk class. This approach has significant advantages for multi-risk bandits because it supports the evaluation of multiple risks simultaneously.

### 4.1 Median of Empirical Quantiles

**Definition 4.1.** Let $X_1, X_2, \cdots, X_n$ be independent and identically distributed random variables with a cumulative distribution function (CDF) $F(x)$. Divide the $n$ samples into $k$ groups of size $N = \lfloor n/k \rfloor$, indexed by $j \in \{1, \cdots, k\}$. If $n$ is not divisible by $k$, the remaining $r = n - kN$

samples are dropped. The empirical CDF and quantile function of the $j$th group is defined as

$$\hat{F}_{N,j}(x) := \frac{1}{N}\sum_{i=1}^{N}\mathbb{I}[X_{i,j} < x], \ \hat{F}_{N,j}^{-1}(y) := \inf\{x : \hat{F}_j(x) \geq y\}, \ y \in [0,1] \tag{5}$$

The median of empirical quantiles (MoEQ) and its empirical CDF are defined as

$$\hat{F}_{\text{med},N}^{-1}(y) := \underset{j\in\{1,\cdots,k\}}{\text{median}} \hat{F}_{N,j}^{-1}(y), \ \hat{F}_{\text{med},N}(x) := \frac{1}{N}\sum_{i=1}^{N}\mathbb{I}[\hat{F}_{\text{med},N}^{-1}(i/N) < x]. \tag{6}$$

*Remark* 4.2 (Validity of MoEQ). We would like to note that $\hat{F}_{\text{med},N}^{-1}$ and $\hat{F}_{\text{med},N}$ are valid quantile function and CDF (See Lemma E.1). Furthermore, $\hat{F}_{\text{med},N}$ also can be represented as the median of CDFs, i.e., $\hat{F}_{\text{med},N}(x) = \text{median}_{j\in\{1,\cdots,k\}}\hat{F}_{N,j}(x)$ (See Lemma E.2). Hence, this MoEQ is in the domain of general risk measure $\rho$, in other words, $\rho(\hat{F}_{\text{med},N})$ is well defined.

While MoEQ offers robustness by aggregating empirical quantiles and reducing the influence of outliers in heavy-tailed distributions, it has a computational drawback, as mentioned in Bhatt et al. [30], that updating MoEQ requires $k$ new samples for the $k$ groups. This dependency makes updates infeasible when fewer than $k$ samples are collected, as balanced group sizes cannot be maintained. To address this issue, we propose a bootstrap resampling.

## 4.2 Bootstrap Resampling for Efficient Updates

Suppose we have $n$ data points divided into $k$ groups, with each group ideally containing $N = \lfloor n/k \rfloor$ samples. If $n$ is not divisible by $k$, the remainder $r = n - kN$ creates imbalance across groups. To address this, two bootstrap resampling approaches can be employed to ensure uniform group sizes while maintaining statistical properties.

**Partial Bootstrap Augmentation.** The first approach, named partial bootstrap augmentation, handles the imbalance by assigning the $r$ remaining samples to $r$ groups, making them contain $N+1$ samples. For the remaining $k - r$ groups, one additional sample is drawn with replacement to bring them to $N+1$ samples. After resampling, the empirical quantile function for each group, $\hat{F}_{N+1,j}^{PB,-1}(y)$, is computed, and the median of these quantiles across all groups is obtained as

$$\hat{F}_{\text{med},N+1}^{PB,-1}(y) := \underset{j\in\{1,\cdots,k\}}{\text{median}} \hat{F}_{N+1,j}^{PB,-1}(y). \tag{7}$$

**Full Bootstrap Augmentation.** The second approach, named full bootstrap, creates all groups by sampling $N$ points with replacement from the original $n$ samples. This method eliminates the need for handling remainders and ensures uniform group sizes across all $k$ groups. For $k$ bootstrap datasets, the median of the quantile functions is defined as

$$\hat{F}_{\text{med},N}^{FB,-1}(y) := \underset{j\in\{1,\cdots,k\}}{\text{median}} \hat{F}_{N,j}^{FB,-1}(y), \tag{8}$$

In summary, we introduce two approaches for balancing group sizes: partial bootstrap augmentation redistributes remainders and resamples for balance, while full bootstrap creates uniform groups by resampling with replacement. Partial bootstrap is preferred as it requires only one bootstrap sample per group, whereas full bootstrap demands a larger resample size to ensure convergence rates, which will be analyzed in the following section.

## 4.3 Deviation Inequality of MoEQs

To analyze the deviation inequalities for MoEQ, partial bootstrap MoEQ (PB-MoEQ), and full bootstrap MoEQ (FB-MoEQ), we assume the stability of the underlying distribution. Specifically, the empirical measure $\hat{\mu}_n$, constructed from $n$ i.i.d. samples, concentrates around the true distribution $\mu$ under a given distance metric $\mathbb{D}$. This stability assumption, formalized below, provides the basis for deriving deviation bounds for the three methods.

**Assumption 4.3** (Stability). Assume that $\mu \in \mathcal{H}(p)$ satisfies the following concentration inequality. Specifically, there exists a distance $\mathbb{D}$ and constant $C_p > 0$, which is only dependent on $p$, such that for any $n \in \mathbb{N}$ and any $x > 0$,

$$\mathbb{P}\left(\mathbb{D}(\hat{\mu}_n, \mu) > x\right) \leq C_p/(n^{p-1}x^p). \tag{9}$$

See Appendix D for examples of distributions and metrics that satisfy this assumption.

This assumption enables us to quantify the deviation bounds for MoEQ under heavy-tailed settings.

**Theorem 4.4.** *Let Assumptions 3.1 and 4.3 hold for some $p \in (1, 2]$, and let $F(x)$ denote the CDF of the distribution with $n$ i.i.d. samples. Consider the MoEQ estimator $\hat{F}_{\mathrm{med},N}^{-1}$ and its empirical distribution $\hat{F}_{\mathrm{med},N}$. Define $k = \lfloor 8 \ln(e^{1/8}/\delta) \rfloor$ and $N = \lfloor n/k \rfloor$. Then, with probability at least $1 - \delta$,*

$$\mathbb{D}\left(\hat{F}_{\mathrm{med},N}, F\right) \leq \beta_{p,n}(\delta), \quad \left|\rho(\hat{F}_{\mathrm{med},N}) - \rho(F)\right| \leq \kappa \beta_{p,n}(\delta), \tag{10}$$

*where $\beta_{p,n}(\delta) := (4C_p \nu_p)^{\frac{1}{p}} \left(16 \ln \left(e^{1/8}/\delta\right)/n\right)^{1-1/p}$.*

Proof can be found in Appendix E.1.1. Theorem 4.4 holds for any distance $\mathbb{D}$ satisfying Assumption 4.3. Importantly, the application to the $W_1$ under heavy-tailed distributions is a notable distinction, as shown in the following corollary.

**Corollary 4.5** (Confidence Intervals for MoEQ Variants). *Under the same assumptions as Theorem 4.4, the following confidence intervals hold with probability at least $1 - \delta$:*

$$W_1\left(\hat{F}, F\right) \leq \beta(\delta), \quad \left|\rho(\hat{F}) - \rho(F)\right| \leq \kappa \beta(\delta), \tag{11}$$

*where $\hat{F}$ and $\beta(\delta)$ depend on the variant as follows.*

- *For MoEQ, $\hat{F} = \hat{F}_{\mathrm{med},N}$, $\beta(\delta) = \beta_{p,n}(\delta)$ defined in Theorem 4.4.*

- *For PB-MoEQ, $\hat{F} = \hat{F}_{\mathrm{med},N+1}^{PB}$, $\beta(\delta) = \beta'_{p,n}(\delta) = (C'_p/C_p)^{1/p} \beta_{p,n}(\delta)$, where $C'_p = 2^{2p} + 2^p C_p$.*

- *For FB-MoEQ, $\hat{F} = \hat{F}_{\mathrm{med},N}^{FB}$, $\beta(\delta) = \beta''_{p,n}(\delta) = 2(8C_p \nu_p)^{1/p}/n^{1-1/p}$, assuming $N \geq \ln(32n)/2 \cdot \left(128n^2/(8C_p)^{1/p}\right)^2$ and $k \geq 8\ln(1/\delta)$.*

Proof can be found in Appendix E.1.2. Our results derive deviation inequalities and confidence intervals for $W_1$ under heavy-tailed distributions, covering $1 < p \leq 2$ without boundedness or finite variance assumptions. This broadens applicability to the widest class of distributions.

*Remark* 4.6 (Comparisons with Prior Work on Deviation Inequalities of $W_1$). First, bounded distributions have been studied extensively in the context of risk measure aware bandits. Tamkin et al. [21] and Liang and Luo [10] used the Dvoretzky-Kiefer-Wolfowitz (DKW) inequality to derive deviation inequalities for $\ell_\infty$ and $W_1$, establishing these metrics as suitable for Lipschitz risk measures in bounded support settings. However, the DKW inequality is insufficient for unbounded distributions. To address unbounded support, Tan et al. [9] extended the analysis to sub-Gaussian, sub-Exponential, and finite variance $(p = 2)$ distributions, deriving deviation inequalities for $W_1$. Yet, their approach does not cover the cases of infinite variance $(p \in (1, 2])$, leaving a gap in the theoretical understanding of heavy-tailed distributions. Bhatt et al. [30] proposed a truncated empirical distribution (TED) framework for $p \in (1, 2]$ but did not address deviation inequalities for $W_1$. To fill this gap and compare TED with MoEQ, we manually derived deviation inequalities, showing that the bounds of TED scale as $O(n^{\frac{1}{2}\left(1 - \frac{1}{p}\right)})$, which is less tight than MoEQ (see Appendix I). Our analysis extends these results by enabling SRM, DRM, UBSR, and CERM to be applied under heavy-tailed distributions, advancing the theoretical framework for Lipschitz risk measures in broader contexts.

## 5 Risk Measure Agnostic Distributional Bandits with Heavy-Tails

We begin with the single-risk setting, where MoEQ is used to estimate the reward distribution for each action. For an action $a$, the MoEQ at round $t$ is denoted as $\hat{X}_{a,T_a(t-1)}$, based on $T_a(t-1)$ observations, which represents the total number of times action $a$ has been chosen up to round $t - 1$. This single-risk algorithm will be further extended to the multi-risk setting, where multiple risk measures $\varrho$ are estimated for each action.

### 5.1 Risk Measure Agnostic Distributional Lower Confidence Bounds with MoEQ

We present a distributional lower confidence bounds (DistLCB) algorithm. Using the deviation inequality and confidence interval derived for MoEQ in Theorem 4.4, the learner computes the following LCB index for each action:

$$\mathrm{LCB}_a(t) := \rho(\hat{X}_{a,T_a(t-1)}) - \kappa \beta_{p,T_a(t-1)}(\delta_t), \tag{12}$$

where $\rho(\hat{X}_{a,T_a(t-1)})$ is the risk measure applied to the MoEQ estimate, and $\beta_{p,T_a(t-1)}(\delta_t)$ is the confidence width determined by Theorem 4.4. The confidence width depends on the number of observations $T_a(t-1)$, the tail parameter $p$, and the confidence level $\delta_t$. At each time step $t$, the learner selects the action $A_t$ with the lowest LCB index: $A_t = \arg\min_{a \in \mathcal{A}} \mathrm{LCB}_a(t)$. This approach uses MoEQ's robustness to estimate heavy-tailed rewards and balance exploration and exploitation with risk measure aware confidence intervals. Detail algorithm is described in Appendix F.

### 5.2 Multi-Risk DistLCB with MoEQ

The multi-risk setting extends the DistLCB to multi-risk distributional LCB (MR-DistLCB) to handle multiple risk measures $\varrho$ by leveraging the robustness of the MoEQ for all Lpschitz risk measures. For each action $a$, the learner computes $r$-dimensional LCB indices similarly to (12). At each time step $t$, the learner estimates the Pareto optimal set $\hat{\mathcal{P}}_{*,t}$ by checking dominance relationships among actions based on LCB indices. An action is selected uniformly at random from $\hat{\mathcal{P}}_{*,t}$, its reward is observed, and the estimates are updated. This extension is possible because our derived results apply to all Lipschitz risk measures and are the first to address 1-Wasserstein distance under heavy-tailed settings. Detail algorithm is described in Appendix F.

## 6 Regret Analysis

In this section, we analyze the regret bounds of two algorithms: DistLCB and MR-DistLCB. We provide both asymptotic lower bounds and regret bounds for these algorithms under heavy-tailed reward distributions. To analyze asymptotic optimality, we first derive the asymptotic and minimax lower bounds of SRM and DRM.

### 6.1 Lower Bounds of Risk Measure Aware Bandits with Heavy-Tailed Rewards

**Theorem 6.1** (Asymptotic Lower Bound for SRM and DRM). *For any $c \in (0, 1/4)$, let $X_1$ and $X_2$ be Bernoulli random variables supported on $\{0, 1/\gamma\}$, where $\gamma = (2c)^{1/(p-1)}$. Their distributions are defined as $\mu_1 := (1 + c\gamma - \gamma^p)\delta_0 + (\gamma^p - c\gamma)\delta_{1/\gamma}$ and $\mu_2 := (1 - \gamma^p)\delta_0 + \gamma^p \delta_{1/\gamma}$ where $\delta_x$ denotes a point mass at $x$. For this Bernoulli bandit problem, the $p$-th moments satisfy $\nu_p = 1$. Consider SRM and DRM as a risk measure and the sub-optimality gap of the second arm is $\Delta_2^\rho = \rho(X_2) - \rho(X_1)$. Then, for any algorithm satisfying $\mathbb{E}[T_2(n)] = o(n^\alpha)$ for $\alpha > 0$, the regret satisfies $\liminf_{n \to +\infty} \mathcal{R}_n^\rho / \ln(n) \geq \Omega\left(1/(\Delta_2^\rho)^{\frac{1}{p-1}}\right)$.*

*Proof Sketch.* Proof can be found in Appendix G.1.1. To derive the asymptotic lower bounds, we construct two Bernoulli random variables $X_1$ and $X_2$ to satisfy $\nu_p = 1$. By using specific properties of SRM and DRM, we compute the sub-optimality gap $\Delta_2^\rho$ explicitly, showing $\Delta_2^\rho \geq \Omega(c)$ for both SRM and DRM. Finally, by employing the generic lower bound of bandits, as in Bubeck et al. [24], Lattimore and Szepesvári [35], and explicitly deriving the KL divergence $D_{\mathrm{KL}}(X_2, X_1) \leq O(c^{\frac{p}{p-1}})$, we establish the asymptotic regret bound as $\Delta_2^\rho / D_{\mathrm{KL}}(X_2, X_1) \geq \Omega\left(1/(\Delta_2^\rho)^{1/(p-1)}\right)$, demonstrating the dependency on the sub-optimality gap and the heavy-tail parameter $p$. $\qquad\square$

*Remark* 6.2 (Comparison with Prior Work on Asymptotic Lower Bounds). Theorem 6.1 presents a novel asymptotic lower bound for regret in risk measure aware bandit problems under SRM and DRM. By extending the approach of Baudry et al. [22], which builds on the generic lower bound from Lai and Robbins [36], we generalize their CVaR-specific results to SRM and DRM. Additionally, we establish that the dependency of the sub-optimality gap $\Delta_2^\rho$ follows the order $1/(p-1)$, providing a broader theoretical foundation for regret bounds in risk measure aware bandits. Moreover, the dependency on $\Delta$ matches the results for expectation-based bandits with heavy-tailed rewards [24]. This similarity is natural since expectation is an SRM.

**Theorem 6.3** (Minimax Lower Bound for SRM and DRM). *Let $K > 1$ and $n \geq K - 1$. Consider SRM and DRM as a risk measure. For any policy, there exists a $K$-armed Bernoulli bandit problem such that $\mathcal{R}_n^\rho \geq \Omega((K-1)^{1-1/p}n^{1/p})$ holds.*

Proof can be found in Appendix G.1.2. The proof strategy is similar to that of Theorem 6.1. Note that, if we set an expectation as a risk measure, the minimax lower bound of SRM (or DRM) matches

that of expectation-based bandits, scaling as $\Omega(K^{1-1/p}n^{1/p})$ [24]. Furthermore, this result is also consistent with the minimax lower bound derived in Bhatt et al. [30].

*Remark* 6.4 (Limitations). A limitation of Theorem 6.1 and 6.3 lies in its focus on SRM and DRM. The proof leverages specific properties of SRM and DRM, making it inapplicable to other risk measures such as UBSR and CERM. As a result, this work does not provide a generalized lower bound for all Lipschitz risk measures. Extending the analysis to include a broader class of risk measures remains an open problem and an important direction for future research.

## 6.2 Optimality of DistLCB

To establish the optimality of the DistLCB algorithm, we analyze its regret bounds under heavy-tailed reward distributions and Lipschitz risk measures.

**Theorem 6.5.** *Consider a $K$-armed stochastic bandit problem with a $\kappa$ Lipschitz risk measure $\rho$ w.r.t. $W_1$ under Assumption 3.1. At each time $t$, define the number of groups in the MoEQ as $k_t = \lfloor 8\ln(e^{1/8}t^4) \rfloor$. Then, the expected regret $\mathcal{R}_n^\rho$ of the DistLCB algorithm has the following bounds,*

$$\text{Gap-Dependent Bound:} \quad O\left( \sum_{a:\Delta_a^\rho > 0} \kappa^{\frac{p}{p-1}} \left( \frac{C_p\nu_p}{\Delta_a^\rho} \right)^{\frac{1}{p-1}} \ln(n) \right)$$

$$\text{Gap-Independent Bound:} \quad O\left( \kappa(C_p\nu_p)^{\frac{1}{p}} \left( K\ln(n) \right)^{\frac{p-1}{p}} n^{\frac{1}{p}} \right)$$

Proof can be found in Appendix G.2. The proof can be done by applying analysis techniques of UCB [1, 24] to the confidence interval of MoEQ. This theorem establishes that the DistLCB achieves asymptotic optimality, with its gap-dependent regret matching the lower bound of Theorem 6.1, scaling as $O\left( 1/(\Delta_a^\rho)^{1/(p-1)} \right)$. Furthermore, the algorithm achieves near-minimax optimality up to a logarithmic factor of $n$.

*Remark* 6.6 (Universality of DistLCB Regret Bounds). This result demonstrates the universality of the regret bounds achieved by the DistLCB across various settings for Lipschitz risk measures and heavy-tailed rewards. For $p = 2$ case that includes bounded support, sub-Gaussian, and sub-Exponential distributions, the regret bounds recover all previous results on expectations [24–26, 28, 29], CVaR-based bandits [2, 7], and Lipschitz risk bandits [8, 9, 30]. This consistency highlights its ability to unify existing analyses across different risk measures and distributional assumptions. For $1 < p < 2$, which have infinite variance, the minimax regret bound aligns with the results of Bhatt et al. [30] for Lipschitz risk bandits under heavy-tailed distributions. Additionally, the gap-dependent regret bound presented here complements existing studies by providing refined guarantees for Lipschitz risk measures in heavy-tailed settings.

## 6.3 Regret Bounds of MR-DistLCB

**Theorem 6.7.** *Consider a $K$-armed stochastic bandit problem with $r$ Lipschitz risk measures $\varrho$ with a vector of Lipschitz constants, $\mathcal{K}$, under Assumption 3.1. At each time $t$, let $k_t = \lfloor 8\ln(e^{1/8}Krt^4) \rfloor$. Then, the regret $\mathcal{R}_n^{\text{Pareto}}$ satisfies the following bounds,*

$$\text{Gap-Dependent Bound:} \quad O\left( \sum_{a \in \mathcal{A} \setminus \mathcal{P}_*} \frac{\kappa_*(C_p\nu_p)^{\frac{1}{p-1}} \ln(Krn)}{(\Delta_a^{\text{Pareto}})^{\frac{1}{p-1}}} \right)$$

$$\text{Gap-Independent Bound:} \quad O\left( \kappa_*(C_p\nu_p)^{\frac{1}{p}} \left( K\ln(Krn) \right)^{\frac{p-1}{p}} n^{\frac{1}{p}} \right),$$

*where $\kappa_* := \max_{d \in [r]} \kappa_d$.*

Proof is provided in Appendix G.3. Since MoEQ constructs confidence intervals for all Lipschitz risk measures, it naturally extends to multi-risk settings. The adaptability of MoEQ to various Lipschitz risk measures builds upon techniques from Drugan and Nowe [31]. For $1 < p \leq 2$, our results provide the first regret bounds for multi-objective bandits under heavy-tailed rewards and MR-DistLCB enables decision-making that considers diverse perspectives of reward distributions.

*Remark* 6.8 (Comparisons with Multi-Objective Bandits). For $p = 2$, the gap-dependent and gap-independent bounds align with Drugan and Nowe [31] and the extended results of Xu and Klabjan [34], matching the lower bounds $\Omega(\ln(n))$ and $\Omega(\sqrt{n})$, respectively.

# 7 Numerical Experiments

**Setup.** We test our methods in synthetic and real-world multi-risk bandit settings with heavy-tailed rewards. We consider two settings: (1) Real-world portfolio selection, using daily returns from the top 20 S&P 500 stocks; and (2) Synthetic 20-armed Pareto bandits. In the real-world case, we evaluate 3-, 6-, and 9-risk settings with CVaR, Wang [11], and CERM. Tail indices estimated via Hill's method range from $p = 1.5$ to $4.69$, and we fix $p = 1.5$ for all algorithms. In the synthetic case, we vary $p \in \{1.01, 1.2, 1.5\}$ to study the effect of tail heaviness, with $p = 1.2$ used as the baseline heavy-tail configuration. All experiments run for 10,000 steps and are repeated with 20 random seeds. Additional setup details are provided in Appendix H.

**Comparison Methods.** We compare against three baselines: (1) **MR Trunc**, a multi-risk adaptation of the truncated empirical distribution (TED) method [30]; (2) **MR LCB**, a multi-risk extension of LCB [9], incorporating ideas from the multi-objective setting of [31]; and (3) **SR DistLCB**, a single-risk version of our method used to assess the necessity of multi-risk exploration.

**Real-World Results.** As shown in Figure 2, MR DistLCB consistently outperforms all baselines across 3-, 6-, and 9-risk settings. This reflects its use of tight, distributionally robust confidence bounds and risk measure aware exploration. MR Trunc performs slightly worse due to its simplified truncation strategy (see Appendix I). This supports the theoretical advantage of MR DistLCB in multi-risk settings, particularly as complexity increases. As noted in Remark 4.6, baseline methods suffer from weaker deviation bounds, which slow convergence for distortion-based risks like Wang and CERM. In contrast, MR DistLCB maintains stable risk estimation as the number of risks grows. MR LCB performs the worst among multi-risk methods, as its sub-Gaussian assumptions lead to unstable or miscalibrated decisions. SR DistLCB performs poorly in the 3-risk case where the Pareto set is small, often failing to identify Pareto-optimal arms. Although its performance improves in 6- and 9-risk settings as the Pareto set expands, it still fails to match MR DistLCB. This confirms that single-risk methods do not suffice for multi-risk optimization.

**Synthetic Results.** Figure 3 compares cumulative Pareto regret across 3-, 6-, and 9-risk configurations under different tail indices $p$. MR-DistLCB consistently achieves the lowest cumulative regret in all settings, while MR Trunc and MR LCB exhibit higher regret and larger variance. The MoEQ-based aggregation enables robust estimation even when reward distributions have infinite variance, whereas truncation-based or sub-Gaussian methods degrade significantly under heavier tails.

Across tail indices $p = 1.01, 1.2, 1.5$, MR-DistLCB maintains the lowest mean regret and smallest standard deviation. For extremely heavy-tailed rewards ($p = 1.01$), its performance advantage becomes most pronounced, highlighting the effectiveness of the exponential-type deviation bound in controlling uncertainty. As $p$ increases and the tails become lighter, all methods exhibit reduced variability, yet MR-DistLCB preserves its lead, demonstrating robustness to varying tail heaviness. MR Trunc remains limited by its conservative cutoff rule, and MR LCB continues to misestimate uncertainty under non-sub-Gaussian noise.

These results collectively confirm that MR-DistLCB achieves strong empirical robustness across diverse tail regimes. Even in the extreme heavy-tail setting ($p \approx 1$), it achieves substantially lower and more stable regret, consistent with the theoretical rates derived in Section 6. This stability across risk complexity and tail indices underscores the method's practical applicability in real-world heavy-tailed domains such as finance and risk-aware decision-making.

# 8 Conclusion

We introduce a Pareto-optimal multi-risk bandit framework under heavy-tailed rewards and derive deviation inequalities for the 1-Wasserstein distance, which enables confidence intervals for Lipschitz risk measures and the first lower bounds for risk measure aware bandits. The proposed MR-DistLCB algorithm achieves minimax-optimal regret bounds and outperforms truncation-based methods in real-world experiments. These results highlight the necessity of specialized multi-risk frameworks for decision-making under uncertainty. Although truncation-based methods show limitations in multi-risk settings, since we do not establish lower bounds for truncation-based LCBs, the possibility

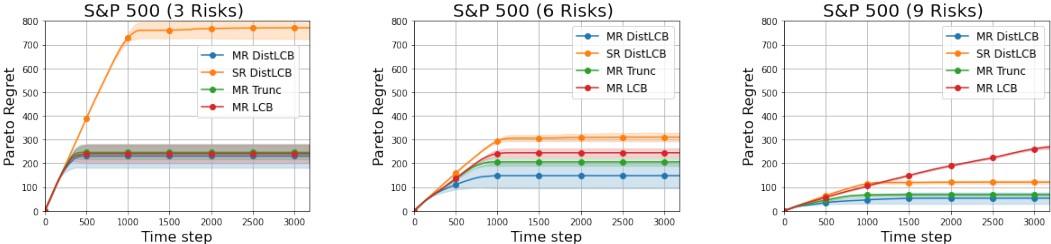

Figure 2: Cumulative regret on the real-world S&P 500 dataset. Solid/dotted lines show averages; shaded areas denote standard deviations.

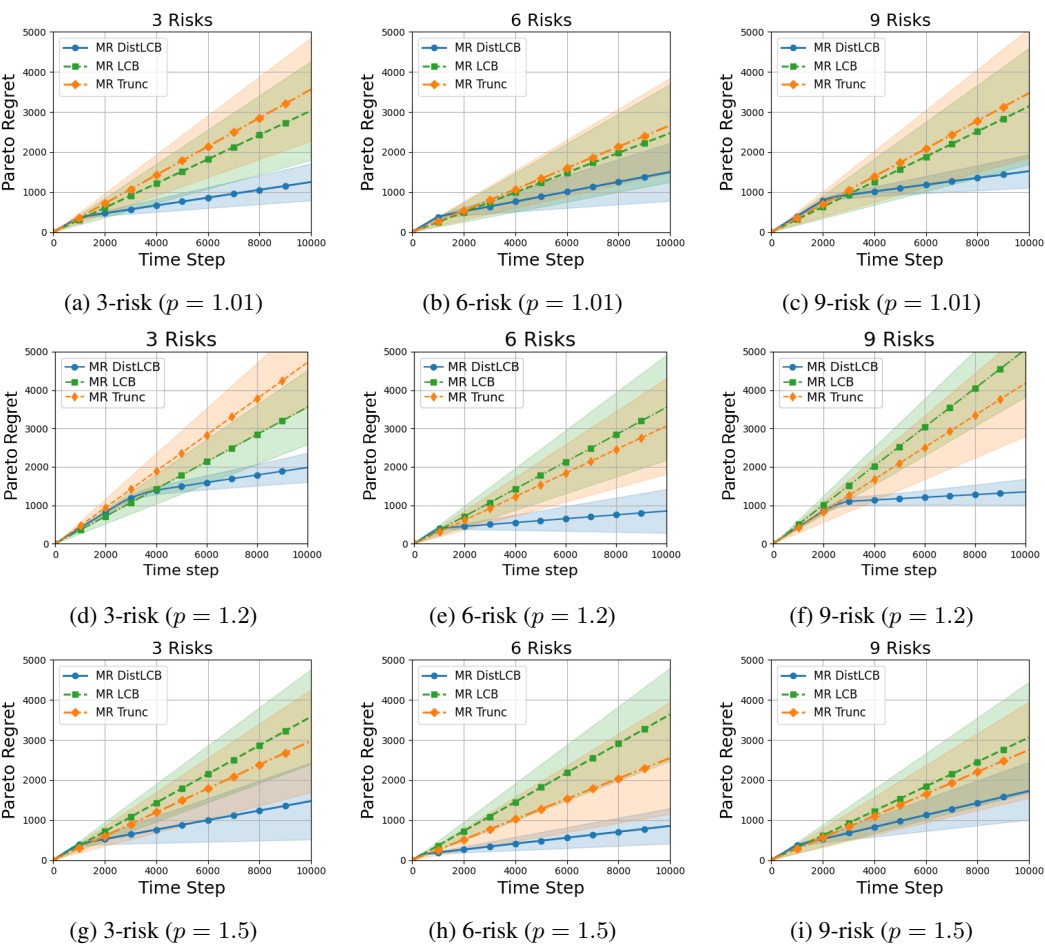

Figure 3: Cumulative regret on the synthetic 20-armed Pareto bandit across different numbers of risk measures (3, 6, 9) and tail indices $p$. Solid and dotted lines represent mean regret across runs, while shaded regions denote one standard deviation. As the tail index decreases (heavier tails), MR DistLCB maintains lower and more stable regret compared to MR Trunc and MR LCB.

of improvement remains open. Future research could investigate whether adaptive truncation or alternative refinements can enhance confidence intervals for SRM and DRM, potentially addressing the observed limitations.

## Acknowledgments

This work was partly supported by the National Research Foundation of Korea (NRF) grant funded by the Korea government (MSIT) (No.RS-2022-NR068754, 25%) and (No.RS-2023-00211357, Smart Assembler: Robot Active Learning for Unseen Parts Assembly, 25%) and (No.RS-2024-00410082, 25%) and (No. NRF-2022M3J6A1063595), and also supported by Institute of Information & communications Technology Planning & Evaluation(IITP) grant funded by the Korea government (MSIT)(No.2022-0-00612, 25%), and partly supported by a Korea University Grant (K2512961).

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

# A  Notations, Definitions, and Assumptions

- $p \in (1, 2]$: Order of the bounded moment
- $\nu_p$: Constant bound of the moment
- $\mathcal{H}(p) := \{\mu \in \mathcal{L}_1 : \exists \nu_p \text{ s.t. } \mathbb{E}_{X \sim \mu}[|X|^p] \leq \nu_p\}$
- $\mu$: Probability measure
- $\hat{\mu}_n$: Empirical measure with $n$ samples
- $\mathcal{L}_1$: Space of probability measure with finite expectation.
- $X_i$: $i$th sample
- $X_{(i)}$: $i$th smallest sample
- $F(x)$ and $F^{-1}(t)$: CDF and Quantile
- $\hat{F}_n(x)$ and $\hat{F}_n^{-1}(x)$: Empirical CDF and quantile with $n$ samples
- $\hat{F}_n^B(x)$ and $\hat{F}_n^{B,-1}(x)$: Empirical CDF and quantile constructed by $n$ bootstrap-resampled points
- $\hat{F}_{\mathrm{med},n}^{-1}$ and $\hat{F}_{\mathrm{med},n}$: Median of empirical quantiles and its corresponding CDF.
- $\hat{F}_{\mathrm{med},n}^{B,-1}$ and $\hat{F}_{\mathrm{med},n}^{B}$: Median of empirical quantiles and its corresponding CDF constructed by partially bootstraped samples.
- $\hat{F}_{\mathrm{med},n}^{FB,-1}$ and $\hat{F}_{\mathrm{med},n}^{FB}$: Median of empirical quantiles and its corresponding CDF constructed by fully bootstraped samples.
- $\mathbb{D}(\mu, \mu')$: Distance between probability measures in $\mathcal{L}_1$
- $W_1$: 1-Wasserstein distance
- $\rho(X)$: Lipschitz risk measure
- $\kappa$: Lipschitz constant of Lipschitz risk measure
- $\mathcal{A}$: Set of actions
- $a \in \mathcal{A}$: Action
- $t$ and $n$: Time step and total rounds (or total number of samples)
- $A_t$: The action selected at time $t$
- $T_a(t)$: The selection count for action $a$ over $t$ steps
- $X_{a,t}$: Random reward of action $a$ at time step $t$
- $\rho_a := \rho(X_a)$: Risk measure of action $a$
- $\hat{X}_{a,n}$: The median of empirical quantiles for action $a$
- $\rho(\hat{X}_{a,n})$: Estimated risk measure for action $a$ based on $n$ samples.
- $\rho(\hat{X}_{a,T_a(t)})$: Estimated risk measure for action $a$ at time $t$, where $T_a(t)$ is the number of times action $a$ has been selected up to time $t$.
- $a_* := \arg\min_{a \in \mathcal{A}} \rho(X_a)$: Optimal action that maximizes the given risk $\rho$.
- $\Delta_a^\rho := \rho(X_a) - \rho(X_{a_*})$: Sub-optimality gap given risk $\rho$
- $\mathcal{R}_n^\rho := \mathbb{E}\left[\sum_{t=1}^n \Delta_{A_t}^\rho\right]$: Cumulative regret
- $\boldsymbol{\varrho}(X_a) := [\rho_1(X_a), \rho_2(X_a), \cdots, \rho_r(X_a)]^\mathsf{T}$: $r$ dimensional risk vector
- $v \prec u$: A vector $u$ dominates a vector $v$
- $v \not\prec u$: A vector $u$ does not dominates a vector $v$. There exists at least one dimension $d$ such that $v_d \geq u_d$ holds.
- $\mathcal{P}_* := \{a \in \mathcal{A} | \forall a' \in \mathcal{A}/\{a\}, \quad \boldsymbol{\varrho}(X_a) \not\prec \boldsymbol{\varrho}(X_{a'})\}$: Pareto optimal actions that are not dominated by any other action across all risk measures.
- $\Delta_a^{\mathrm{Pareto}} := \inf_{\epsilon \geq 0}\{\epsilon \mid \boldsymbol{\varrho}(X_a) + \epsilon \mathbf{1} \parallel \boldsymbol{\varrho}(X_{a'}), \forall a' \in \mathcal{P}_*\}$: Pareto sub-optimality gap
- $\mathcal{R}_n^{\mathrm{Pareto}} := \mathbb{E}\left[\sum_{t=1}^n \Delta_{A_t}^{\mathrm{Pareto}}\right]$: Cumulative Pareto regret

# B  Detailed Comparison of Prior Work

This appendix presents a structured comparison of prior risk measure aware bandit methods across risk types, reward distributions, and theoretical guarantees.

**risk measure aware Bandits with Light-Tailed Rewards.**    Numerous studies have analyzed bandits under *light-tailed rewards* for specific risk measures, including CVaR [2, 3] or mean-variance[6, 7]. Especially, CVaR-based bandits have garnered significant attention among risk measures. Galichet et al. [2] achieved $O(\ln(n))$ regret bounds for CVaR-based bandits with bounded rewards, and Vakili and Zhao [7], Maillard [3] achieved similar bounds under sub-Gaussian assumptions. For bounded support distributions, Tamkin et al. [21] and Baudry et al. [22] provided regret bounds with detailed dependency on the sub-optimality gap, achieving problem-dependent bounds of $O(\sum_a \log(n)/\Delta_a^{\mathrm{CVaR}})$ and problem-independent bounds of $O(\sqrt{Kn\log(n)})$, where $\Delta_a^{\mathrm{CVaR}}$ is the sub-optimality gap under CVaR. Recently, Tan and Weng [23] extended these results to sub-Gaussian reward settings for stochastic and adversarial CVaR bandits. These works significantly advance optimization for specific single risk measures but do not generalize to broader risk classes or heavy-tailed settings.

**risk measure aware Bandits for Lipschitz Risk Measures.**    Beyond single risk measures, researchers have developed bandit algorithms for Lipschitz risk measures [8–10], which is a broad class including CVaR, SRM, DRM, UBSR, and CERM. These risk measures satisfy Lipschitz continuity with respect to distribution distances, allowing algorithms to address multiple risk metrics in this class, simultaneously. Several studies have explored Lipschitz risk measures in risk measure aware bandit settings. Cassel et al. [8] considered sub-Gaussian rewards and achieved regret bounds of $O(\ln(n))$. A. and Bhat [37], Tan et al. [9] expanded the scope further to sub-Exponential and finite variance distributions, with regret bounds of $O(\ln(n))$ for sub-Gaussian rewards, $O(\sqrt{n})$ for sub-Exponential distributions, and $O(n^{p/(p+1)})$ for finite variance distributions ($2 < p$), where $p$ is the order of finite moment. Liang and Luo [10] extended this result to locally Lipschitz risk measures under bounded support rewards. However, none of these works address distributions with infinite variance, leaving heavy-tailed rewards unexplored.

**risk measure aware Bandits with Heavy-Tailed Rewards.**    Most studies on heavy-tailed distributions have focused on specific risk settings, such as maximizing expected rewards [24–29] or optimizing CVaR [4, 5], while research on a broad class of risk measures under heavy-tailed assumptions remains scarce. To the best of our knowledge, Bhatt et al. [30] is the only study that addresses Lipschitz risk class under heavy-tailed reward distributions with infinite variance. Bhatt et al. [30] analyzed bandits under piecewise-stationary regimes with heavy-tailed rewards, achieving regret bounds of $\tilde{O}(\sqrt{Kn})$ for gap-dependent cases and $\tilde{O}(K^{1-1/p}n^{1/p})$ for worst-case scenarios, where $p \in (1, 2]$ is the largest order of the finite moment.[3] They also proposed a truncated empirical distribution method applicable to the Lipschitz risk class. However, since their analysis considers non-stationary distributions, it does not provide tight regret bounds for multi-risk settings in stationary distributions.

**Multi-Risk Bandits.**    Multi-risk bandits extend single-risk settings by optimizing multiple risk measures simultaneously, often through Pareto-optimality. Existing works [31–34] have focused on Pareto regret minimization but are limited to light-tailed distributions and do not address broader risk classes like Lipschitz risk measures. Efforts to combine multi-risk settings with heavy-tailed rewards or generalize to diverse risk measures remain unexplored.

**Lower Bounds of risk measure aware Bandits.**    Lower regret bounds in risk measure aware bandit settings depend on the chosen risk measure and the reward distribution assumptions. For CVaR-based bandits under the sub-Gaussian assumption, Baudry et al. [22] established an asymptotic lower bound of $\Omega(\sum_a \Delta_a^{\mathrm{CVaR}} \log(n)/K_{\inf}^\alpha(\nu_a, c^\alpha))$. However, no asymptotic lower bounds exist for Lipschitz risk measures, even under sub-Gaussian distributions, and remain unexplored for heavy-tailed rewards.

---

[3]Bhatt et al. [30] originally considered piecewise-stationary reward settings, where the regret bounds are $\tilde{O}(\sqrt{MKn})$ for gap-dependent cases and $\tilde{O}((MK)^{1-1/p}n^{1/p})$ for worst-case scenarios. Here, $M$ represents the number of distribution changes. Setting $M = 1$ corresponds to stationary heavy-tailed reward settings, which is the focus of this paper.

Establishing the dependency of asymptotic lower bounds on the sub-optimality gap (e.g., $\Delta_k^{\mathrm{CVaR}}$) for generalized risk classes and heavy-tailed settings remains a critical open problem. We would like to note that the minimax lower bound on general risk measure aware bandits are derived as $\Omega(K^{1-1/p}n^{1/p})$ in Bhatt et al. [30].

## C   Summary of Lipschitz Risk Measure

The following well-known risk measures satisfy Lipschitz continuity defined in Definition 3.2. Note that we generally follow the definition of risk measures in Tan et al. [9] and Liang and Luo [10].

- **Conditional Value-at-Risk (CVaR):**
  - Definition: CVaR at level $\alpha \in (0,1)$ is defined as
  $$\mathrm{CVaR}_\alpha(X) = \frac{1}{1-\alpha} \int_\alpha^1 F^{-1}(t)\,dt, \tag{13}$$
  - Lipschitz Constant: $(1-\alpha)^{-1}$, where $\alpha$ is the risk level.
  - Distance Metric: $W_1$.
  - Condition on Lipschitzness: None explicitly required beyond the definition.
- **Spectral Risk Measure (SRM):**
  - Definition: A spectral risk measure is defined as
  $$\mathrm{SRM}(X) = \int_0^1 \phi(t) F^{-1}(t)\,dt, \tag{14}$$
  where $\phi(t)$ is the risk spectrum satisfying $\phi(t) \geq 0$ and $\int_0^1 \phi(t)\,dt = 1$ and increasing function.
  - Lipschitz Constant: $\sup_{\beta \in [0,1]} \phi(\beta)$, where $\phi(\beta)$ is the risk spectrum.
  - Distance Metric: $W_1$.
  - Condition on Lipschitzness: $\phi(\beta)$ must be bounded.
- **Distortion Risk Measure (DRM)**
  - Definition: A distortion risk measure is defined as:
  $$\mathrm{DRM}(F) = \int_0^\infty g(1 - F(x))\,dx, \tag{15}$$
  where $g : [0,1] \to [0,1]$ is a continuous, concave, and increasing distortion function satisfying $g(0) = 0$ and $g(1) = 1$. We further assume that $g$ is strictly increasing $g'(x) > 0$, thus, $\inf_{x \in [0,1]} g'(x) > 0$ holds.
  - Lipschitz Constant: $\|g'\|_\infty$, where $\|g'\|_\infty$ is the supremum of the derivative $g'$ over its domain $[0,1]$.
  - Distance Metric: $W_1$ (1-Wasserstein distance).
  - Condition on Lipschitzness: The derivative $g'(t)$ of the distortion function must be bounded, i.e., $g'(t) \leq \|g'\|_\infty$ for all $t \in [0,1]$. To prove the Lipschitz property of DRM under the Wasserstein distance $W_1$, we apply the Mean Value Theorem for the distortion function $g$, yielding $|g(1 - F(x)) - g(1 - G(x))| \leq \|g'\|_\infty |F(x) - G(x)|$. Integrating over the domain gives:
  $$|\mathrm{DRM}(F) - \mathrm{DRM}(G)| \leq \|g'\|_\infty W_1(F, G). \tag{16}$$
  Thus, the Lipschitz constant of DRM with respect to $W_1$ is $\|g'\|_\infty$.
- **Utility-Based Shortfall Risk (UBSR):**
  - Definition: UBSR is defined as
  $$\mathrm{UBSR}(X) = \inf\{m \in \mathbb{R} : \mathbb{E}[l(X - m)] \leq \eta\}, \tag{17}$$
  where $l$ is a utility function and $\eta > 0$ is a threshold parameter.
  - Lipschitz Constant: $\frac{K}{k}$, where $K$ is the Lipschitz constant of the utility function $l$, and $k$ is its strong convexity parameter.
  - Distance Metric: $W_1$.

- Condition on Lipschitzness: The utility function $l$ must be $K$-Lipschitz and satisfy the strong convexity property $l(x_2) \geq l(x_1) + k(x_2 - x_1)$ for $x_2 \geq x_1$.

- **Certainty Equivalent (CE) Risk Measure:**
  - Definition: The Certainty Equivalent (CE) risk measure is defined as

  $$\mathrm{CE}(F) = u^{-1}\left(\int_{-\infty}^{\infty} u(x)\, dF(x)\right),\qquad(18)$$

  where $u(x)$ is a continuous, strictly increasing, and differentiable utility function, and $u^{-1}$ is its inverse.
  - Lipschitz Constant:

  $$\frac{\|u'\|_\infty}{\inf_{x \in \text{range}(u)} u'(x)},\qquad(19)$$

  where $\|u'\|_\infty$ is the supremum of the derivative $u'(x)$, and $\inf_{x \in \text{range}(u)} u'(x)$ is the minimum derivative over the range of $u$.
  - Distance Metric: $W_1$.
  - Condition on Lipschitzness: The utility function $u(x)$ must be continuously differentiable, with $u'(x)$ bounded above and away from zero, i.e., $0 < \inf_{x \in \text{range}(u)} u'(x) \leq u'(x) \leq \|u'\|_\infty < \infty$.

# D   On the Validity of Assumption 4.3

In this section, we provide additional justification and concrete examples supporting Assumption 4.3, which postulates a polynomial concentration inequality for the empirical distribution under a given distance metric. The validity of this assumption depends on both the underlying distributional class and the choice of distance metric. Below, we demonstrate that the assumption is satisfied in a range of settings, including both heavy-tailed and light-tailed regimes.

**(i) $\ell_\infty$ distance under heavy-tailed distributions** ($1 < p$).   By the Dvoretzky-Kiefer-Wolfowitz (DKW) inequality [38], for the empirical distribution function $\hat{F}$ of $n$ i.i.d. samples and the true CDF $F$, we have:

$$\mathbb{P}\left(\sup_u |\hat{F}(u) - F(u)| > x\right) \leq 2\exp(-2nx^2).$$

Since the exponential bound decays faster than any polynomial, there exists a constant $C_p$ such that:

$$\mathbb{P}\left(\sup_u |\hat{F}(u) - F(u)| > x\right) \leq \frac{C_p \nu_p}{n^{p-1} x^p}, \quad \text{for all } n \in \mathbb{N}, x > 0.$$

**(ii) $W_1$ distance under heavy-tailed distributions** ($1 < p \leq 2$).   Dedecker and Merlevède [39, Theorem 2.1] provide the following deviation bound for the 1-Wasserstein distance between $\hat{F}$ and $F$ under finite $p$-th moment conditions:

$$\mathbb{P}(W_1(\hat{F}, F) > x) \leq \frac{C_p \nu_p}{n^{p-1} x^p}.$$

**(iii) $W_1$ distance under bounded, sub-Gaussian, or sub-Exponential distributions.**   Prashanth and Bhat [40] show that for light-tailed distributions, deviation bounds for the Wasserstein distance admit exponential tails:

$$\mathbb{P}(W_1(\hat{F}, F) > x) \leq O\left(\exp(-nx^\beta) + \exp(-nx^2)\right),$$

for some $\beta > 0$. Since the exponential decay dominates polynomial tails, this again satisfies Assumption 4.3 with appropriate constants.

**(iv) $\ell_1$ distance under bounded support.** Liang and Luo [10] establish the following concentration bound:

$$\mathbb{P}(\ell_1(\hat{F}, F) > x) \leq O\left(\exp\left(-c \cdot \frac{n}{e}\left(\frac{x-32}{8(b-a)}\right)^2\right)\right),$$

where $[a, b]$ denotes the support. Again, exponential decay implies the existence of $C_p$ satisfying Assumption 4.3.

These results confirm that Assumption 4.3 is satisfied under commonly used distance metrics across both heavy-tailed and light-tailed settings. In the final version of the paper, we will include these clarifications along with appropriate citations to ensure completeness and transparency.

# E  Median of Empirical Quantiles

## E.1  Median of Empirical Quantiles

**Lemma E.1.** $\hat{F}_{\mathrm{med},N}^{-1}(y)$ and $\hat{F}_{\mathrm{med},N}(x)$ are a valid quantile and cumulative distribution function, respectively.

*Proof.* First, the domain of $\hat{F}_{\mathrm{med},N}^{-1}(y)$ is clearly $y \in [0, 1]$. Second, we prove the non-decreasing property. It is equivalent to show that, for any $y_1 \leq y_2$, $\hat{F}_{\mathrm{med},N}^{-1}(y_1) \leq \hat{F}_{\mathrm{med},N}^{-1}(y_2)$. If $k$ is odd, say $k = 2v + 1$ for some integer $v$, then, by definition of the median, at least $v$ number of $F_{N,j}^{-1}(y_1)$ are greater than $\hat{F}_{\mathrm{med},N}^{-1}(y_1)$ and at least $v$ number of $F_{N,j}^{-1}(y_1)$ are less than $F_{N,j}^{-1}(y_1)$. Since $F_{N,j}^{-1}(y)$ is non-decreasing, we have, for all $j$, $F_{N,j}^{-1}(y_1) \leq F_{N,j}^{-1}(y_2)$. Hence, any empirical CDF $F_{N,j}^{-1}(y_1)$ that was greater than $\hat{F}_{\mathrm{med},N}^{-1}(y_1)$ satisfies the same condition at $y_2$, i.e., $\hat{F}_{\mathrm{med},N}^{-1}(y_1) \leq F_{N,j}^{-1}(y_1) \leq F_{N,j}^{-1}(y_2)$. So, at least $v$ number of empirical CDFs are still greater than $\hat{F}_{\mathrm{med},N}^{-1}(y_1)$ at $y_2$. In this regard, $\hat{F}_{\mathrm{med},N}^{-1}(y_2)$ cannot be less than $\hat{F}_{\mathrm{med},N}^{-1}(y_1)$, i.e., $\hat{F}_{\mathrm{med},N}^{-1}(y_1) \leq \hat{F}_{\mathrm{med},N}^{-1}(y_2)$. For even number $k = 2v$, the same argument also holds. Third, by definition of $\hat{F}_{\mathrm{med},N}(x)$, its quantile function is $\hat{F}_{\mathrm{med},N}^{-1}(y)$. $\square$

**Lemma E.2.** For all $x \in \mathbb{R}$, $\hat{F}_{\mathrm{med},N}(x) = \underset{j \in \{1,2,\cdots,k\}}{\mathrm{median}}\left(\hat{F}_{N,j}(x)\right)$ holds.

*Proof.*

$$\hat{F}_{\mathrm{med},N}(x) = \frac{1}{N}\sum_{i=1}^{N}\mathbb{I}\left[\hat{F}_{\mathrm{med},N}^{-1}(i/N) < x\right] = \frac{1}{N}\sum_{i=1}^{N}\mathbb{I}\left[\mathrm{median}_j(X_{(i),j}) < x\right] \tag{20}$$

$$= \frac{1}{N}\sum_{i=1}^{N}\mathrm{median}_j(\mathbb{I}\left[X_{(i),j} < x\right]) = \mathrm{median}_j\left(\frac{1}{N}\sum_{i=1}^{N}\mathbb{I}\left[X_{(i),j} < x\right]\right) \tag{21}$$

$$= \underset{j \in \{1,2,\cdots,k\}}{\mathrm{median}}\left(\hat{F}_{N,j}(x)\right) \tag{22}$$

$\square$

### E.1.1  Proof of Theorem 4.4

*Proof.* First, we would like to note that, due to convexity of $\mathbb{D}(\cdot, F)$, we have

$$\mathbb{D}(\hat{F}_{\mathrm{med},N}, F) \leq \underset{\ell \in [k]}{\mathrm{median}}\left(\mathbb{D}(\hat{F}_{N,\ell}, F)\right), \tag{23}$$

where detail proof can be found in Merkle [41]. Then, the similar technique in Bubeck et al. [24] is applied. Let $x > 0$ and $Y_\ell := \mathbb{I}[W_1(\hat{F}_{N,\ell}, F) > x]$ for $\ell \in [1, 2, 3, \cdots, k]$. Then, from the Assumption 4.3, we have,

$$q := \mathbb{P}(Y_\ell = 1) = \mathbb{P}(\mathbb{D}(\hat{F}_{N,\ell}, F) > x) \leq \frac{C_p \nu_p}{x^p N^{p-1}} \tag{24}$$

Note that for

$$x = \frac{(4C_p \nu_p)^{1/p}}{N^{1-1/p}} \tag{25}$$

we have $q \leq 1/4$. Then, by using Hoeffding's inequality for the tail of a binomial distribution, we get

$$\mathbb{P}\left(\mathbb{D}\left(\hat{F}_{\mathrm{med},N}, F\right) \geq x\right) \leq \mathbb{P}\left(\underset{\ell \in [k]}{\mathrm{median}}\left(\mathbb{D}(\hat{F}_{N,\ell}, F)\right) > x\right) \tag{26}$$

$$= \mathbb{P}\left(\sum_{\ell=1}^{k} Y_\ell > \frac{k}{2}\right) \leq \exp\left(-2k(1/2 - q)^2\right) \tag{27}$$

$$\leq \exp\left(-k/8\right) \leq \delta. \tag{28}$$

where the last inequality holds since $k = \lfloor 8\ln(e^{1/8}/\delta) \rfloor > 8\ln(1/\delta)$. Hence, we have $N = \lfloor n/k \rfloor > \frac{n}{16\ln(e^{1/8}/\delta)}$. Consequently, with probability at least $1 - \delta$,

$$\mathbb{D}\left(\hat{F}_{\mathrm{med},N}, F\right) \leq x = \frac{(4C_p \nu_p)^{1/p}}{N^{1-1/p}} \leq (4C_p \nu_p)^{1/p} \left(\frac{16\ln(e^{1/8}/\delta)}{n}\right)^{1-1/p}. \tag{29}$$

$\square$

### E.1.2 Proof of Corollary 4.5

To handle bootstrap resampling, we first derive the following two lemmas.

**Lemma E.3** (Deviation of PB-MoEQ). *Under the same conditions as Theorem 4.4 and the 1-Wasserstein distance $W_1$, let $\hat{F}_{N+1,j}^{B}$ be one sample bootstrapped empirical CDF of $\hat{F}_{N,j}$. Then, for any fixed $x > 0$,*

$$\mathbb{P}\left(W_1(\hat{F}_{N+1,j}^{B}, \hat{F}_{N,j}) > x\right) \leq \frac{2^{p+1}\nu_p}{N^{p-1}x^p} \tag{30}$$

*Proof.* To quantify the difference between the empirical distributions before and after adding a bootstrap-resampled sample, we compute the 1-Wasserstein distance:

$$W_1(\hat{F}_{N,j}, \hat{F}_{N+1,j}^{B}) = \int_{\mathbb{R}} \left|\hat{F}_{N,j}(x) - \hat{F}_{N+1,j}^{B}(x)\right| dx. \tag{31}$$

Then, the difference between the two distributions is:

$$\hat{F}_{N+1,j}^{B}(x) - \hat{F}_{N,j}(x) = \frac{1}{N+1}\mathbb{I}[X_{\mathrm{Boot}} \leq x] - \frac{1}{N(N+1)}\sum_{i=1}^{N}\mathbb{I}[X_i \leq x]. \tag{32}$$

where $X_{\mathrm{Boot}}$ is a bootstrapped sample. Then, reorganizing, this becomes:

$$\hat{F}_{N+1,j}^{B}(x) - \hat{F}_{N,j}(x) = \frac{1}{N+1}\left(\mathbb{I}[X_{\mathrm{Boot}} \leq x] - \hat{F}_{N,j}(x)\right). \tag{33}$$

Thus, the Wasserstein distance is expressed as:

$$W_1(\hat{F}_{N,j}, \hat{F}_{N+1,j}^{B}) = \frac{1}{N+1}\int_{\mathbb{R}}\left|\mathbb{I}[X_{\mathrm{Boot}} \leq x] - \hat{F}_{N,j}(x)\right| dx \leq \frac{1}{N+1}2\max_{i \in [N]}|X_i|. \tag{34}$$

where the last inequality holds since the gap between the indicator and empirical CDF is clearly zero outside of $[-\max_i|X_i|, \max_i|X_i|]$. Now, for the deviation probability,

$$\mathbb{P}\left(W_1(\hat{F}_{N+1,j}^{B}, \hat{F}_{N,j}) > x\right) \leq \mathbb{P}\left(\frac{1}{N+1}2\max_{i \in [N]}|X_i| > x\right) \tag{35}$$

$$\leq N\mathbb{P}\left(|X_i| > \frac{(N+1)x}{2}\right) \leq \frac{2^p \nu_p N}{(N+1)^p x^p} \leq \frac{2^p \nu_p}{N^{p-1}x^p} \tag{36}$$

$\square$

**Lemma E.4** (Deviation of FB-MoEQ). *Under the same conditions as Theorem 4.4 and the 1-Wasserstein distance $W_1$, let $\hat{F}_{N,\ell}^{FB}$ be the $\ell$th full bootstrap empirical CDF, and let $\hat{F}_n$ be the empirical CDF obtained by original dataset. Then, for any $N$, $t > 0$, and any fixed $x > 0$,*

$$\mathbb{P}\left(W_1(\hat{F}_{N,\ell}^{FB}, \hat{F}_n) > x\right) \leq 2n\exp\left(-2N\left(\frac{x}{4nt}\right)^2\right) + \frac{2n\nu_p}{t^p}. \tag{37}$$

*Proof.* Consider the conditional probability:

$$\mathbb{P}(W_1(\hat{F}_{N,\ell}^{FB}, \hat{F}_n) > x) = \mathbb{E}_{X_1,\cdots,X_n}\left[\mathbb{P}(W_1(\hat{F}_{N,\ell}^{FB}, \hat{F}_n) > x \mid X_1, \cdots, X_n)\right]. \tag{38}$$

Then, given original datasets, the Wasserstein distance can be bounded by:

$$W_1(\hat{F}_{N,\ell}, \hat{F}_n) = \int_{-\infty}^{\infty} |\hat{F}_{N,\ell}(x) - \hat{F}_n(x)|dx = \int_{-\max_i |X_i|}^{\max_i |X_i|} |\hat{F}_{N,\ell}(x) - \hat{F}_n(x)|dx \tag{39}$$

$$\leq \int_{-\max_i |X_i|}^{\max_i |X_i|} \sum_{i=1}^{n} \left|\frac{k_i}{N} - \frac{1}{n}\right| \mathbb{I}[X_i < x]dx \leq \sum_{i=1}^{n} \left|\frac{k_i}{N} - \frac{1}{n}\right| \cdot 2\max_i |X_i|, \tag{40}$$

where $k_i$ is a random variable that indicates the number of times $X_i$ is selected in the bootstrap resample. Given the original samples, uniform resampling with replacement is equivalent to sampling $N$ independent categorical random variables with equal probability $1/n$. Thus, we have:

$$\mathbb{P}(W_1(\hat{F}_{N,\ell}, \hat{F}_n) > x \mid X_1, \cdots, X_n) \leq \mathbb{P}\left(\sum_{i=1}^{n} \left|\frac{k_i}{N} - \frac{1}{n}\right| > \frac{x}{2\max_i |X_i|} \Bigg| X_1, \cdots, X_n\right) \tag{41}$$

$$\leq \sum_{i=1}^{n} \mathbb{P}\left(\left|\frac{k_i}{N} - \frac{1}{n}\right| > \frac{x}{2n\max_i |X_i|} \Bigg| X_1, \cdots, X_n\right). \tag{42}$$

Note that marginal probability of $k_i$ is a binomial distribution with $N$ and $1/n$. Using the Hoeffding inequality for the deviation of a binomial variable:

$$\mathbb{P}\left(W_1(\hat{F}_{N,\ell}, \hat{F}_n) > x \mid X_1, \cdots, X_n\right) \leq 2n \exp\left(-2N\left(\frac{x}{4n\max_i |X_i|}\right)^2\right). \tag{43}$$

Now, consider the total probability, for $t > 0$,

$$\mathbb{P}(W_1(\hat{F}_{N,\ell}, \hat{F}_n) > x/2) = \mathbb{P}(W_1(\hat{F}_{N,\ell}, \hat{F}_n) > x/2 \mid \max_i |X_i| \leq t)\mathbb{P}(\max_i |X_i| \leq t) \tag{44}$$

$$+ \mathbb{P}(W_1(\hat{F}_{N,\ell}, \hat{F}_n) > x/2 \mid \max_i |X_i| > t)\mathbb{P}(\max_i |X_i| > t). \tag{45}$$

For the first term:

$$\mathbb{P}(W_1(\hat{F}_{N,\ell}, \hat{F}_n) > x \mid \max_i |X_i| \leq t) \leq 2n \exp\left(-2N\left(\frac{x}{4nt}\right)^2\right). \tag{46}$$

For the second term, using the tail bound on $\max_i |X_i|$:

$$\mathbb{P}(\max_i |X_i| > t) \leq \frac{2n\nu_p}{t^p}. \tag{47}$$

Combining these, we have,

$$\mathbb{P}(W_1(\hat{F}_{N,\ell}, \hat{F}_n) > x) \leq 2n \exp\left(-2N\left(\frac{x}{4nt}\right)^2\right) + \frac{2n\nu_p}{t^p}. \tag{48}$$

$\square$

By using the deviation inequalities of bootstrap resampling cases, we now prove Corollary 4.5.

*Proof of Corollary 4.5.* **Proof of MoEQ:** First, for the case of MoEQ, let us verify that the 1-Wasserstein distance satisfies Assumption 4.3 to apply Theorem 4.4. From Theorem 2.1 in Dedecker and Merlevède [39], for an empirical distribution of heavy-tailed distribution with $p > 1$, we have:

$$\mathbb{P}(W_1(\hat{F}_{N,\ell}, F) > x) \leq \frac{C_p\nu_p}{N^{p-1}x^p}. \tag{49}$$

This implies that $W_1$ satisfies Assumption 4.3. By applying Theorem 4.4, we know that $\beta(\delta) = \beta_{p,n}(\delta) = (4C_p\nu_p)^{1/p}(16\ln(e^{1/8}/\delta)/n)^{1-1/p}$. Hence, the proof of the first case is completed.

**Proof of PB-MoEQ:** For the case of PB-MoEQ, it follows directly from the proof of the Lemma E.3. By the triangle inequality:

$$\mathbb{P}\left(W_1\left(\hat{F}_{N+1,j}^B, F\right) > x\right) \leq \mathbb{P}\left(W_1\left(\hat{F}_{N+1,j}^B, \hat{F}_{N,j}\right) > \frac{x}{2}\right) + \mathbb{P}\left(W_1\left(\hat{F}_{N,j}, F\right) > \frac{x}{2}\right) \tag{50}$$

$$\leq \frac{2^{2p}\nu_p}{N^{p-1}x^p} + \frac{2^p C_p\nu_p}{N^{p-1}x^p} = \frac{C_p'\nu_p}{N^{p-1}x^p}, \tag{51}$$

where $C'_p := 2^{2p} + 2^p C_p$. From this fact, we can apply Theorem 4.4. From Theorem 4.4, we get:

$$W_1\left(\hat{F}^B_{\text{med},N+1}, F\right) \le (4C'_p \nu_p)^{1/p} \left(\frac{16 \ln(e^{1/8}/\delta)}{n}\right)^{1-1/p} = \beta_{p,n}(\delta)'. \tag{52}$$

This completes the proof.

**Proof of FB-MoEQ** For the case of FB-MoEQ, it is enough to prove the following probability bound,

$$\mathbb{P}\left(W_1\left(\hat{F}^{FB}_{N,j}, F\right) > x\right) \le \mathbb{P}\left(W_1\left(\hat{F}^{FB}_{N,j}, \hat{F}_n\right) > x/2\right) + \mathbb{P}\left(W_1\left(\hat{F}_n, F\right) > x/2\right) \tag{53}$$

Then, for the first term, by using the Lemma E.4, we get,

$$\mathbb{P}(W_1(\hat{F}^{FB}_{N,\ell}, \hat{F}_n) > x/2) \le 2n \exp\left(-2N\left(\frac{x}{8nt}\right)^2\right) + \frac{2n\nu_p}{t^p} \tag{54}$$

$$\le 2n \exp\left(-2N\left(\frac{x}{256n^{1+\frac{1}{p}}\nu_p^{\frac{1}{p}}}\right)^2\right) + \frac{1}{16} \text{ by choosing } t = 32n^{1/p}\nu_p^{1/p} \tag{55}$$

For the second term, again, from Theorem 2.1. in Dedecker and Merlevède [39], we have,

$$\mathbb{P}(W_1(\hat{F}_n, F) > x/2) \le \frac{2^p C_p \nu_p}{n^{p-1} x^p} \tag{56}$$

Finally, by combining two results, we get,

$$q := \mathbb{P}\left(W_1\left(\hat{F}^{FB}_{N,j}, F\right) > x\right) \le 2n \exp\left(-2N\left(\frac{x}{256n^{1+\frac{1}{p}}\nu_p^{\frac{1}{p}}}\right)^2\right) + \frac{1}{16} + \frac{2^p C_p \nu_p}{n^{p-1} x^p} \tag{57}$$

Note that for

$$x = \frac{(8 \cdot 2^p \cdot C_p \nu_p)^{1/p}}{n^{1-1/p}}, N \ge \frac{\ln(32n)}{2}\left(\frac{128n^2}{(8C_p)^{1/p}}\right)^2, \tag{58}$$

we get $q \le 1/4$. Applying Hoeffding's inequality for the median, the probability bound is established, leading to:

$$W_1\left(\hat{F}^{FB}_{\text{med},N}, F\right) \le \frac{2(8C_p \nu_p)^{1/p}}{n^{1-1/p}} = \beta_{p,n}(\delta)''. \tag{59}$$

Consequently, this completes the proof. $\qquad\square$

# F    Algoirthmic Details

## F.1    Distributional LCB

The Distributional LCB algorithm extends the classical lower confidence bound approach to risk measure aware settings by leveraging a Lipschitz-continuous risk measure. At each round, the algorithm constructs a confidence interval for the estimated risk of each arm using the median-of-empirical-quantiles (MoEQ) estimator, and selects the arm with the lowest lower confidence bound. This design enables robust decision-making under heavy-tailed reward distributions.

---

**Algorithm 1** Distributional LCB

---

1: **Input:** $\mathcal{A}$: Set of actions, $K$: Number of actions, $(\rho, \kappa)$: A Lipschitz risk measure with Lipschitz constant, $n$: Total time steps, $p$: The order of a bounded moment, $\nu_p$: The bound of the $p$th moment, $C_p$: Scaling factor of deviation bound, $k$: Number of empirical quantiles
2: **Output:** Optimal action $\hat{a}_*$
3: Initialize $T_a \leftarrow 0$ for all $a \in \mathcal{A}$, $\hat{X}_{a,0} \leftarrow 0$ for all $a \in \mathcal{A}$, $LCB(a) \leftarrow 0$ for all $a \in \mathcal{A}$
4: **for all** $a \in \mathcal{A}$ **do**
5:     Pull arm $a$ and observe reward $X_a$, update $\hat{X}_{a,1}$, and increment $T_a \leftarrow T_a + 1$
6: **end for**
7: **for** $t = K + 1$ to $n$ **do**
8:     **for all** $a \in \mathcal{A}$ **do**
9:         Compute $\beta_{a,t-1}$ based on Corollary 4.5
10:        $LCB(a) \leftarrow \rho(\hat{X}_{a,T_a}) - \kappa\beta_{a,T_a}$
11:    **end for**
12:    Select action $A_t \leftarrow \arg\min_{a \in \mathcal{A}} LCB(a)$
13:    Observe reward $X_{A_t}$ and update $T_{A_t} \leftarrow T_{A_t} + 1$
14:    Update $\hat{X}_{A_t,T_{A_t}} \leftarrow \text{MoEQ}(\{X_{A_t,1}, \ldots, X_{A_t,T_{A_t}}\})$
15: **end for**
16: **Return:** $\hat{a}_* \leftarrow \arg\min_{a \in \mathcal{A}} \rho(\hat{X}_{a,T_a})$

---

### F.2 Median of Empirical Quantiles with Bootstrap Resampling

The Median of Empirical Quantiles (MoEQ) estimator is used to construct robust estimates of the reward distribution under heavy-tailed noise. The data is partitioned into multiple blocks, and the empirical quantiles of each block are computed. The final estimate is obtained by taking the median across blocks at each quantile level. To ensure balanced partitioning, bootstrap resampling is applied if the sample size is not divisible by the number of blocks.

---

**Algorithm 2** Median of Empirical Quantiles with Bootstrap Resampling

---

1: **Input:** $X_1, X_2, \ldots, X_n$: $n$ Samples, $k$: Number of empirical quantiles (or groups)
2: **Output:** The computed median quantiles $\{\hat{F}_M^{-1}(j/N) : j = 1, 2, \ldots, N\}$
3: If $n \mod k \neq 0$, bootstrap-resample from $X_1, X_2, \ldots, X_n$ to ensure $n = kN$.
4: Compute block size: $N \leftarrow \lfloor n/k \rfloor$.
5: Divide the data into $k$ groups, each of size $N$:

$$\text{Group}_1 = \{X_1, \ldots, X_N\}, \text{Group}_2 = \{X_{N+1}, \ldots, X_{2N}\}, \ldots, \text{Group}_k = \{X_{(k-1)N+1}, \ldots, X_{kN}\}. \tag{60}$$

6: Sort each group: For each $\text{Group}_i$, sort the observations to compute order statistics:

$$X_{(1),i} \leq X_{(2),i} \leq \cdots \leq X_{(N),i}. \tag{61}$$

7: Compute the median of quantiles: For each quantile level $\frac{j}{N}$, compute:

$$\hat{F}_M^{-1}(j/N) \leftarrow \text{Median}\big(\{X_{(j),1}, X_{(j),2}, \ldots, X_{(j),k}\}\big). \tag{62}$$

---

### F.3 Multi-Risk Distributional LCB

The Multi-Risk Distributional LCB algorithm generalizes the risk measure aware exploration to settings involving multiple Lipschitz risk measures. For each arm and risk type, the algorithm computes lower confidence bounds and identifies a Pareto-optimal set of actions that are non-dominated across all considered risks. An action is then sampled uniformly from this Pareto set, balancing risk-sensitive exploration under multiple objectives. This approach is particularly useful in applications where trade-offs between different types of risks must be considered simultaneously.

---

**Algorithm 3** Multi-Risk Distributional LCB

---

1: **Input:** $\mathcal{A}$: Set of actions, $K$: The number of actions, $\{(\rho_1, \kappa_1), (\rho_2, \kappa_2), \ldots, (\rho_r, \kappa_r)\}$: Set of $r$ Lipschtiz risk measures with Lipschtiz constants, $n$: Total time steps, $p$: The order of a bounded moment, $\nu_p$: The bound of the $p$th moment, $C_p$: Scaling factor, $k$: The number of empirical quantiles

2: **Ouput:** Pareto optimal actions $\hat{\mathcal{P}}_{*,n}$

3: Initialize $\hat{\mathcal{P}}_{*,0} \leftarrow \emptyset$

4: Initialize $T_a \leftarrow 0$ for all $a \in \mathcal{A}$

5: Initialize $\hat{X}_{a,0} \leftarrow 0$ for all $a \in \mathcal{A}$

6: Initialize $LCB(a, d) \leftarrow 0$ for all $a \in \mathcal{A}, \ d \in (1, 2, \cdots, r)$

7: **for all** $a \in \mathcal{A}$ **do**

8:     Pull arm $a$ and observe reward $X_a$

9:     Update $\hat{X}_{a,1}$ and increment $T_a \leftarrow T_a + 1$

10: **end for**

11: **for** $t = K + 1$ to $n$ **do**

12:     **for all** $a \in \mathcal{A}$ **do**

13:         Compute $\beta_{a,T_a}$ based on Corollary 4.5

14:         **for** $d = 1$ to $r$ **do**

15:             $LCB(a, d) = \rho_d(\hat{X}_{a,T_a}) - \kappa_d \beta_{a,T_a}$

16:         **end for**

17:     **end for**

18:     **for all** $a \in \mathcal{A}$ **do**

19:         Add $a$ to $\hat{\mathcal{P}}_{*,t}$ if $\forall a' \in \mathcal{A} \setminus \{a\}, LCB(a') \nprec LCB(a)$ {Pareto dominance check}

20:     **end for**

21:     Sample $A_t \sim \text{Uniform}(\hat{\mathcal{P}}_{*,t})$ {Select action}

22:     Observe reward $X_{A_t}$ and update $T_{A_t} \leftarrow T_{A_t} + 1$ for selected action $A_t$

23:     Update $\hat{X}_{A_t,T_{A_t}} \leftarrow \text{MoEQ}(\{X_{A_t,1}, \cdots, X_{A_t,T_{A_t}}\})$ {Update quantiles with all observed rewards of action $A_t$}

24: **end for**

---

# G   Regret Analysis

## G.1   Lower Bounds of risk measure aware Bandits with Heavy-Tailed Rewards

### G.1.1   Asymptotic Lower Bounds

*Proof.* The proof will be done by using the generic lower bound (See Chapter 16 in Lattimore and Szepesvári [35]). Especially, we apply the similar techniques in Bubeck et al. [24] for SRM and DRM. we construct two Bernoulli random variable $X_1$ and $X_2$ on the set $\{0, 1/\gamma\}$, where $\gamma := (2c)^{1/(p-1)}$ and $c \in (0, 1/4)$. Specifically, define the probability

$$\mu_1 = (1 + c\gamma - \gamma^p)\delta_0 + (\gamma^p - c\gamma)\delta_{1/\gamma}. \tag{63}$$

and

$$\mu_2 = (1 - \gamma^p)\delta_0 + \gamma^p \delta_{1/\gamma}, \tag{64}$$

Here, $\delta_x$ denotes a point mass at $x$. Furthermore, we can check the $p$th moment of Bernoulli distributions. Under the distribution $\mu_1$, the random variable $X_1$ takes the value 0 with probability $(1 + c\gamma - \gamma^p)$ and $\frac{1}{\gamma}$ with probability $\gamma^p - c\gamma$. Therefore,

$$\mathbb{E}[|X_1|^p] = 0^p \cdot (1 + c\gamma - \gamma^p) + \left(\frac{1}{\gamma}\right)^p \cdot (\gamma^p - c\gamma) = 1 - c\gamma^{1-p} = 1/2. \tag{65}$$

For $X_2$, we have

$$\mathbb{E}[|X_2|^p] = 0^p \cdot (1 - \gamma^p) + \left(\frac{1}{\gamma}\right)^p \cdot \gamma^p = 1. \tag{66}$$

Hence, for this Bernoulli bandit problem, $\nu_p = 1$ for any choice of $c$.

Then, let us compute a *spectral risk measure* and *distortion risk measure* for these two actions. First, for SRM, we get the following risk values. For $X_1$, we have $p_1 = \gamma^p - c\gamma$ and $x_1 = \frac{1}{\gamma}$. Thus

$$\rho(X_1) = \frac{1}{\gamma} \int_{1-\gamma^p+c\gamma}^{1} \phi(\alpha)\,d\alpha. \tag{67}$$

For $X_2$, we have $p_2 = \gamma^p$ and $x_2 = \frac{1}{\gamma}$. Hence

$$\rho(X_2) = \frac{1}{\gamma} \int_{1-\gamma^p}^{1} \phi(\alpha)\,d\alpha \tag{68}$$

Then, the sub-optimality gap of $X_2$ and $X_1$ becomes,

$$\Delta_2^{\rho} := \rho(X_2) - \rho(X_1) = \frac{1}{\gamma} \int_{1-\gamma^p}^{1-\gamma^p+c\gamma} \phi(\alpha)\,d\alpha \geq \phi(1-\gamma^p)c \geq \phi(1-(1/2)^{p/(p-1)})c = D_{\phi,p}c, \tag{69}$$

where $D_{\phi,p} := \phi(1-(1/2)^{p/(p-1)})$.

Next, for DRM, we get the following DRM values. For $X_1$, we have $p_1 = \gamma^p - c\gamma$ and $x_1 = \frac{1}{\gamma}$. Thus

$$\rho(X_1) = \frac{1}{\gamma}\, g(\gamma^p - c\gamma). \tag{70}$$

For $X_2$, we have

$$\rho(X_2) = \frac{1}{\gamma}\, g(\gamma^p). \tag{71}$$

Similar to SRM, the sub-optimality gap becomes,

$$\Delta_2^{\rho} := \rho(X_2) - \rho(X_1) = \frac{1}{\gamma}\, g(\gamma^p) - \frac{1}{\gamma}\, g(\gamma^p - c\gamma) = g'(\gamma^p - tc\gamma)c \geq D_{g,p}c, \tag{72}$$

where there exists $t \in [0,1]$ such that the last equality holds by the mean value theorem and $D_{g,p} := \inf_x g'(x)$.

From the asymptotic regret lower bound for two-armed bandit problems, as in [24, 35], the regret lower bound becomes

$$\liminf_{n \to +\infty} \frac{\mathcal{R}_n}{\ln(n)} \geq \Delta_2^{\rho} \cdot \frac{1}{D_{\mathrm{KL}}(X_2, X_1)} \geq \frac{\Delta_2^{\rho} \cdot p_2(1-p_2)}{(p_2 - p_1)^2} = \frac{\Delta_2^{\rho} \cdot \gamma^p(1-\gamma^p)}{c^2\gamma^2} \tag{73}$$

$$= \frac{\Delta_2^{\rho} \cdot (2c)^{\frac{p}{p-1}}(1-\gamma^p)}{c^{\frac{2p}{p-1}} 2^{\frac{2}{p-1}}} \geq \frac{\Delta_2^{\rho}(1-(1/2)^{\frac{p}{p-1}})}{c^{\frac{p}{p-1}} 2^{\frac{2-p}{p-1}}} \geq \Omega\left(\frac{1}{(\Delta_2^{\rho})^{\frac{1}{p-1}}}\right). \tag{74}$$

$$\square$$

### G.1.2 Minimax Lower Bound

*Proof.* We prove the theorem using the minimax lower bound technique for $K$-armed bandits. We define two different bandit problems. For the first bandit problem, define $K$ Bernoulli distributions where the first action has the distribution of (63) and other remaining $K-1$ actions have the distribution of (64). Then, the sub-optimality gap of SRM and DRM is

$$\Delta_i^{\rho} \geq D_{\rho,p}c \quad \text{for } i \geq 2, \tag{75}$$

where $D_{\rho,p}$ are constant only dependent on $\rho$ and $p$. To create the second bandit problem, let $i = \arg\min_{j>1} \mathbb{E}_1[T_j(n)]$, where $\mathbb{E}_1$ indicates the expectation over the first bandit problem and $T_j(n)$ denotes the number of times action $j$ is pulled up to time $n$. Since the total number of pulls satisfies $\sum_{j=1}^{K} \mathbb{E}_1[T_j(n)] = n$, it follows that $\mathbb{E}_1[T_i(n)] \leq \frac{n}{K-1}$. Then, for the second bandit problem, we only change the distribution of action $i$ as follows,

$$\mu_i' = \delta_0 \tag{76}$$

where action $i$ has all probability at zero. Here, by construction of both bandit problems, action 1 is optimal under the first bandit problem, while action $i$ is optimal under the second bandit problem. Furthermore, for the second bandit problem, the sub-optimality gap of SRM of the first arm 1 is obtained as

$$\Delta_1^\rho = \rho(X_1') - \rho(X_i') = \frac{1}{\gamma} \int_{1-\gamma^p+c\gamma}^{1} \phi(\alpha)\,d\alpha \tag{77}$$

$$\geq \phi(1-\gamma^p+c\gamma)(\gamma^{p-1}-c) \geq \phi(1-(1/2)^{\frac{2p-1}{p-1}})c = D'_{\phi,p}c, \tag{78}$$

where $X_1'$ and $X_i'$ indicate the Bernoulli distribution for the second problem. For DRM, the gap becomes

$$\Delta_1^\rho = \rho(X_1') - \rho(X_i') = \frac{1}{\gamma}\,g(\gamma^p - c\gamma) - \frac{1}{\gamma}g(0) \geq D'_{g,p}(\gamma^{p-1}-c) = D'_{g,p}c. \tag{79}$$

Furthermore, the KL divergence between $X_i$ and $X_i'$ is obtained as

$$D_{\mathrm{KL}}(X_i, X_i') \leq \frac{(p_1-p_1')^2}{p_1(1-p_1)} = \frac{(\gamma^p - c\gamma)^2}{(\gamma^p - c\gamma)(1-\gamma^p+c\gamma)} = \frac{(2^{\frac{p}{p-1}} - 2^{\frac{1}{p-1}})c^{\frac{p}{p-1}}}{(1 - (2^{\frac{p}{p-1}} - 2^{\frac{1}{p-1}})c^{\frac{p}{p-1}})} \tag{80}$$

$$\leq \frac{(2^{\frac{p}{p-1}} - 2^{\frac{1}{p-1}})c^{\frac{p}{p-1}}}{(1 - (1/2)^{\frac{2p-1}{p-1}})} \tag{81}$$

Then, the regret of the first problem is

$$\mathcal{R}_{n,1}^\rho \geq \sum_{a\neq 1} \Delta_a^\rho \mathbb{E}_1[T_a(n)] \geq \sum_{a\neq 1} \Delta_a^\rho \mathbb{E}_1[T_a(n)\mathbb{I}(T_1(n) \leq n/2)] \tag{82}$$

$$\geq D_{\rho,p}c \sum_{a\neq 1} \mathbb{E}_1[T_a(n)\mathbb{I}(T_1(n) \leq n/2)] \geq \frac{D_{\rho,p}nc}{2}\mathbb{P}_1(T_1(n) \leq n/2) \tag{83}$$

The regret of the second problem is

$$\mathcal{R}_{n,2}^\rho \geq \sum_{a\neq i} \Delta_a^\rho \mathbb{E}_2[T_a(n)] \geq \Delta_1^\rho \mathbb{E}_2[T_1(n)\mathbb{I}(T_1(n) > n/2)] \geq \frac{D'_{\rho,p}nc}{2}\mathbb{P}_2(T_1(n) > n/2) \tag{84}$$

by using the Bretagnolle-Huber inequality, the regret of the two bandit problems satisfies

$$\mathcal{R}_{n,1}^\rho + \mathcal{R}_{n,2}^\rho \geq \frac{\min(D_{\rho,p}, D'_{\rho,p})nc}{4} \exp\left(-\mathbb{E}_1[T_i(n)] \cdot D_{\mathrm{KL}}(X_i, X_i')\right) \tag{85}$$

$$\geq \frac{\min(D_{\rho,p}, D'_{\rho,p})nc}{4} \exp\left(-\frac{E_p nc^{\frac{p}{p-1}}}{K-1}\right). \tag{86}$$

where $E_p := \frac{(2^{\frac{p}{p-1}} - 2^{\frac{1}{p-1}})}{(1-(1/2)^{\frac{2p-1}{p-1}})}$. Consequently, the result is completed by choosing $c = ((K-1)/n)^{1-1/p}$ as follows,

$$\max(\mathcal{R}_{n,1}^\rho, \mathcal{R}_{n,2}^\rho) \geq \frac{\mathcal{R}_{n,1}^\rho + \mathcal{R}_{n,2}^\rho}{2} \geq \Omega\left((K-1)^{1-1/p}n^{1/p}\right). \tag{87}$$

$\square$

### G.2 Optimality of Distributional LCB

*Proof.* **1. Gap-Dependent Bound:** At any time $t$, the event $\{A_t = a\}$ (where action $a$ is chosen at time step $t$) can occur under the following scenarios:

$$\{A_t = a\} \subset \left\{\rho(\hat{X}_{a_*, T_{a_*}(t-1)}) - \kappa(4C_p\nu_p)^{1/p}\left(\frac{8\ln(e^{1/8}t^4)}{T_1(t-1)}\right)^{1-1/p} > \rho(X_{a_*})\right\} \tag{88}$$

$$\cup \left\{\rho(\hat{X}_{a, T_a(t-1)}) \leq \rho(X_a) - \kappa(4C_p\nu_p)^{1/p}\left(\frac{8\ln(e^{1/8}t^4)}{T_a(t-1)}\right)^{1-1/p}\right\} \tag{89}$$

$$\cup \{T_a(t-1) < u\}, \tag{90}$$

where $u = 32(2\kappa)^{\frac{p}{p-1}} \left( \frac{(4C_p\nu_p)^{\frac{1}{p-1}}}{(\Delta_a^\rho)^{\frac{p}{p-1}}} \right) \ln(e^{1/32}n)$. The expected number of times action $a$ is chosen is:

$$\mathbb{E}[T_a(n)] = 1 + \sum_{t=K+1}^{n} \mathbb{P}(A_t = a) \le 1 + u + \sum_{t=u+1}^{n} \mathbb{P}(A_t = a, T_a(t-1) > u) \tag{91}$$

$$\le 1 + u + \sum_{t=u+1}^{n} \mathbb{P}\left( \rho(\hat{X}_{a_*, T_{a_*}(t-1)}) - \kappa(4C_p\nu_p)^{1/p} \left( \frac{8\ln(e^{1/8}t^4)}{T_{a_*}(t-1)} \right)^{1-1/p} > \rho(X_{a_*}) \right) \tag{92}$$

$$+ \sum_{t=u+1}^{n} \mathbb{P}\left( \rho(\hat{X}_{a, T_a(t-1)}) \le \rho(X_a) - \kappa(4C_p\nu_p)^{1/p} \left( \frac{8\ln(e^{1/8}t^4)}{T_a(t-1)} \right)^{1-1/p} \right) \tag{93}$$

$$\le 1 + u + \sum_{t=u+1}^{n} \sum_{s=1}^{t-1} \frac{2}{t^4} \le 1 + u + \sum_{t=u+1}^{n} \frac{2}{t^3} \le u + 2. \tag{94}$$

Substituting $u$, we obtain:

$$\mathbb{E}[T_a(n)] \le 32(2\kappa)^{\frac{p}{p-1}} \left( \frac{(4C_p\nu_p)^{\frac{1}{p-1}}}{(\Delta_a^\rho)^{\frac{p}{p-1}}} \right) \ln(e^{1/32}n) + 2. \tag{95}$$

The regret for action $a$ is:

$$\mathcal{R}_n^\rho = \sum_{a:\Delta_a^\rho > 0} \Delta_a^\rho \mathbb{E}[T_a(n)] \tag{96}$$

$$< \sum_{a:\Delta_a^\rho > 0} 32(2\kappa)^{\frac{p}{p-1}} \left( \frac{4C_p\nu_p}{\Delta_a^\rho} \right)^{\frac{1}{p-1}} \ln(e^{1/32}n) + 2 \sum_{a:\Delta_a^\rho > 0} \Delta_a^\rho. \tag{97}$$

## 2. Gap-Independent Bound:

Define $\Delta = \left( \frac{32K\ln(e^{1/32}n)}{n} \right)^{\frac{p-1}{p}} \kappa(4C_p\nu_p)^{\frac{1}{p}}$. The regret is:

$$\mathcal{R}_n^\rho = \sum_{a:\Delta_a^\rho > 0} \Delta_a^\rho \mathbb{E}[T_a(n)] \tag{98}$$

$$< n\Delta + \sum_{a:\Delta_a^\rho > \Delta} \Delta_a^\rho \mathbb{E}[T_a(n)] \tag{99}$$

$$< n\Delta + \sum_{a:\Delta_a^\rho > \Delta} 32(2\kappa)^{\frac{p}{p-1}} \left( \frac{4C_p\nu_p}{\Delta_a^\rho} \right)^{\frac{1}{p-1}} \ln(e^{1/32}n) + 2 \sum_{a:\Delta_a^\rho > 0} \Delta_a^\rho \tag{100}$$

$$< n\Delta + K32(2\kappa)^{\frac{p}{p-1}} \left( \frac{4C_p\nu_p}{\Delta} \right)^{\frac{1}{p-1}} \ln(e^{1/32}n) + 2 \sum_{a:\Delta_a^\rho > 0} \Delta_a^\rho \tag{101}$$

$$< 4\kappa(4C_p\nu_p)^{\frac{1}{p}} \left( 32K\ln(e^{1/32}n) \right)^{\frac{p-1}{p}} n^{\frac{1}{p}} + 2 \sum_{a:\Delta_a^\rho > 0} \Delta_a^\rho. \tag{102}$$

$$\square$$

### G.3 Regret Bounds of Multi-Risk Distributional LCB

*Proof.* For any action $a \notin \mathcal{P}_*$, we decompose the number of times it is pulled up to time $n$ as

$$T_a(n) = 1 + \sum_{t=K+1}^{n} \mathbb{I}[A_t = a] \le 1 + \ell + \sum_{t=1}^{n} \mathbb{I}[A_t = a, T_a(t-1) \ge \ell]. \tag{103}$$

The second term can be further bounded

$$T_a(n) \le 1 + \ell + \sum_{t=1}^{n} \mathbb{I}\left[ a \in \hat{\mathcal{P}}_{*,t}, T_a(t-1) \ge \ell \right]. \tag{104}$$

From the fact that $a$ is sampled from $\hat{\mathcal{P}}_{*,t}$, this becomes

$$T_a(n) \le 1 + \ell + \sum_{t=1}^{n} \sum_{a_* \in \mathcal{P}_*} \mathbb{I}\left[\boldsymbol{\varrho}(\hat{X}_{a_*,t-1}) - \mathcal{K}\beta_{a_*,t-1} \not\prec \boldsymbol{\varrho}(\hat{X}_{a,t-1}) - \mathcal{K}\beta_{a,t-1}, T_a(t-1) \ge \ell\right]. \quad (105)$$

Next, we use the union bound and consider the following decomposition;

$$\left\{\boldsymbol{\varrho}(\hat{X}_{a_*,s}) - \mathcal{K}\beta_{a_*,s} \not\prec \boldsymbol{\varrho}(\hat{X}_{a,s_a}) - \mathcal{K}\beta_{a,s_a}\right\} \subseteq \quad (106)$$

$$\left\{\boldsymbol{\varrho}(X_a) - \mathcal{K}\beta_{a,s_a} \not\prec \boldsymbol{\varrho}(\hat{X}_{a,s_a})\right\} \cup \left\{\boldsymbol{\varrho}(\hat{X}_{a_*,s}) - \mathcal{K}\beta_{a_*,s} \not\prec \boldsymbol{\varrho}(X_{a_*})\right\} \cup \left\{\boldsymbol{\varrho}(X_{a_*}) \not\prec \boldsymbol{\varrho}(X_a) - 2\mathcal{K}\beta_{a,s_a}\right\}. \quad (107)$$

For $s_a \ge \ell > \frac{(\max_d \kappa_d)(4C_p\nu_p)^{\frac{1}{p-1}} 8\ln(Krn^4)}{(\Delta_a^{\mathrm{Pareto}})^{\frac{p}{p-1}}}$, the third term vanishes

$$\left\{\boldsymbol{\varrho}(X_{a_*}) \not\prec \boldsymbol{\varrho}(X_a) - 2\mathcal{K}\beta_{a,s_a}\right\} = \emptyset. \quad (108)$$

The first two terms can be bounded using concentration inequalities

$$\mathbb{P}\left(\boldsymbol{\varrho}(X_a) - \mathcal{K}\beta_{a,s_a} \not\prec \boldsymbol{\varrho}(\hat{X}_{a,s_a})\right) \le r\frac{1}{Krt^4} = \frac{1}{Kt^4} \quad (109)$$

$$\mathbb{P}\left(\boldsymbol{\varrho}(\hat{X}_{a_*,s}) - \mathcal{K}\beta_{a_*,s} \not\prec \boldsymbol{\varrho}(X_{a_*})\right) \le r\frac{1}{Krt^4} = \frac{1}{Kt^4}. \quad (110)$$

Substituting these bounds into the original sum,

$$\mathbb{E}[T_a(n)] \le 1 + \ell + \sum_{t=1}^{\infty} \sum_{a_* \in \mathcal{P}_*} \sum_{s=1}^{t-1} \sum_{s_a=\ell}^{t-1} \frac{2}{Kt^4} \quad (111)$$

$$\le 1 + \ell + \sum_{t=1}^{\infty} \sum_{a_* \in \mathcal{P}_*} \frac{2}{Kt^2} \quad (112)$$

$$\le \ell + 1 + \frac{\pi^2}{3} \quad (113)$$

Finally, substitute $\ell = \frac{(\max_d \kappa_d)(4C_p\nu_p)^{\frac{1}{p-1}} 8\ln(e^{1/8}Krn^4)}{(\Delta_a^{\mathrm{Pareto}})^{\frac{p}{p-1}}}$, we have

$$\mathbb{E}[T_a(n)] \le \frac{(\max_d \kappa_d)(4C_p\nu_p)^{\frac{1}{p-1}} 8\ln(e^{1/8}Krn^4)}{(\Delta_a^{\mathrm{Pareto}})^{\frac{p}{p-1}}} + 1 + \frac{\pi^2}{3}. \quad (114)$$

Substituting the bound for $\mathbb{E}[T_a(n)]$, we obtain,

$$\mathcal{R}_n^{\mathrm{Pareto}} \le \sum_{a \in \mathcal{A} \setminus \mathcal{P}_*} \frac{(\max_d \kappa_d)(4C_p\nu_p)^{\frac{1}{p-1}} 32\ln(e^{1/32}(Kr)^{1/4}n)}{(\Delta_a^{\mathrm{Pareto}})^{\frac{1}{p-1}}} + \left(1 + \frac{\pi^2}{3}\right) \sum_{a \in \mathcal{A} \setminus \mathcal{P}_*} \Delta_a^{\mathrm{Pareto}}. \quad (115)$$

By repeating the same procedure to compute the worst case bound, the minimax regret bound is obtained

$$\mathcal{R}_n^{\mathrm{Pareto}} \le 2(\max_d \kappa_d)(4C_p\nu_p)^{\frac{1}{p}} \left(32K\ln(e^{1/32}(Kr)^{1/4}n)\right)^{\frac{p-1}{p}} n^{\frac{1}{p}} + \left(1 + \frac{\pi^2}{3}\right) \sum_{a \in \mathcal{A} \setminus \mathcal{P}_*} \Delta_a^{\mathrm{Pareto}}. \quad (116)$$

$\square$

# H   Experimental Details

## H.1   Real-World Experiment Setup

We use daily stock return data from the top 20 S&P 500 companies over 3,184 trading days (2012.05.18–2025.01.14). At each time step, the algorithm selects one asset and observes its daily return as the reward. The task is to identify Pareto-optimal assets across multiple risk measures:

- **3-risk**: $CVaR_{0.75}$, Wang($-1.0$), CERM($\alpha = 1.0$)
- **6-risk**: + $CVaR_{0.9}$, Wang($0.0$), CERM($\alpha = 2.0$)
- **9-risk**: + $CVaR_{0.95}$, Wang($1.0$), CERM($\alpha = 3.0$)

For CERM, $u(x) := -e^{\alpha x}$ is used as a weight function. Tail indices estimated via Hill's estimator range from 1.50 to 4.69. We use $p = 1.5$ conservatively in our algorithm.

The heavy-tailed nature of stock returns is well-documented [13–17]. Using Hill's estimator, we find tail indices in the range $[1.50, 4.69]$, confirming heavy-tailed behavior. We adopt a conservative tail index of $p = 1.5$.

## H.2 Synthetic Experiment Setup

We simulate a 20-armed bandit problem with Pareto-distributed rewards. Arms $a_{18}$, $a_{19}$, and $a_{20}$ belong to the Pareto-optimal set. We vary the number of risks across:

- **3-risk**: $CVaR_{0.9}$, $SRM_1$, $SRM_2$
- **6-risk**: $SRM_1$ to $SRM_6$
- **9-risk**: $CVaR_{0.9}$, $SRM_1$ to $SRM_8$

Each SRM is implemented as a step-function spectral risk measure (SRM) following the axiomatic framework of **?** ]. Given a quantile level $x \in (0, 1]$, the spectrum function $\phi(x)$ is defined as a right-continuous step function that assigns weight $\phi_k$ to quantiles below threshold $\alpha_k$. Formally, the spectrum is evaluated as:

$$\phi(x) = \sum_{k=1}^{K} \phi_k \cdot \mathbb{I}(x > \alpha_k),$$

where $\mathbb{I}(\cdot)$ is the indicator function, and the weights $\{\phi_k\}$ are normalized to satisfy the coherence condition:

$$\sum_{k=1}^{K} \phi_k(1 - \alpha_k) = 1.$$

This construction allows SRMs to be expressed as weighted averages of quantile values, where the shape of the spectrum reflects varying degrees of risk aversion.

We adopt this formulation to flexibly construct a variety of SRMs by adjusting the $\alpha_k$ and $\phi_k$ parameters. This enables controlled evaluation of our algorithms under diverse risk preferences and weight concentrations. The specific parameterizations used in our experiments are listed below:

- $SRM_1$: `alphas` = [0.2, 0.4, 0.6, 0.8], `ranges` = [0.25, 0.5, 0.75, 1.0]
- $SRM_2$: `alphas` = [0.1, 0.2, 0.5, 0.9], `ranges` = [0.2, 0.2, 1.0, 1.6]
- $SRM_3$: `alphas` = [0.5, 0.75, 0.9], `ranges` = [1.0, 1.0, 2.5]
- $SRM_4$: `alphas` = [0.25, 0.5, 0.75], `ranges` = [0.5, 1.0, 0.5]
- $SRM_5$: `alphas` = [0.2, 0.4, 0.6, 0.8, 0.95], `ranges` = [0.0, 0.333, 1.0, 1.0, 4.0]
- $SRM_6$: `alphas` = [0.2, 0.5, 0.8], `ranges` = [0.375, 1.0, 1.0]
- $SRM_7$: `alphas` = [0.15, 0.35, 0.55, 0.75], `ranges` = [0.0, 0.462, 1.0, 1.0]
- $SRM_8$: `alphas` = [0.3, 0.7], `ranges` = [1.0, 1.0]
- $CVaR_{0.9}$: `alpha` = [0.9]

Each SRM captures a different attitude toward risk by shifting the emphasis toward different regions of the tail. Reward means are sampled uniformly over $[0, 1]$ and sorted in ascending order. We run each experiment for 10,000 steps with 20 random seeds. We use a fixed tail index of $p = 1.2$, which simulates a more extreme heavy-tailed condition than observed in the S&P 500 dataset, allowing us to evaluate performance under high-risk environments.

# I  Discussions on Truncated Empirical Distribution

The truncated empirical distribution (TED), introduced by Bhatt et al. [30], is designed to handle heavy-tailed reward distributions by mitigating the impact of extreme outliers. The truncated empirical CDF is defined as

$$\hat{F}_{\text{trunc},n}(x) := \frac{1}{n} \sum_{i=1}^{n} \mathbb{I}[\text{sign}(X_i) \min(|X_i|, i^{\frac{1}{p}}) < x] \tag{117}$$

This truncation limits the contribution of extreme rewards by replacing large values $|X_i|$ with a truncation level $i^{1/p}$, where $p > 1$ is the order of the bounded moment.

Then, the lower confidence bound (LCB) for TED, as formulated by Bhatt et al. [30], is computed as

$$LCB_a(t) = \rho(\hat{F}_{\text{trunc},T_a(t-1)}) - 2q_1 \max\left(A_t, A_t^2\right), \tag{118}$$

where $\rho$ is a Lipschitz risk measure, $T_a(t-1)$ is the number of times action $a$ has been chosen up to time $t-1$, and

$$A_t := \frac{\ln(t)}{3(T_a(t-1))^{1-1/p}} + \frac{\ln(t)}{3(T_a(t-1))^{3/2-1/p}}. \tag{119}$$

Here, $q_1$ is a tunable parameter to adjust the confidence width, defined in Bhatt et al. [30].

We would like to emphasize that confidence interval of the TED in (118) holds for the risk class defined in Bhatt et al. [30] including CVaR. Bhatt et al. [30] derive deviation bounds for the semi-norm, which is equivalent to $\ell_2$ for CVaR, but their framework does not address the Wasserstein distance $W_1$.

In this work, we extend their analysis to the Wasserstein distance $W_1$. Let us first assume that there exist an increasing sequence of threshold $B_n$. While it is $i^{1/p}$ in (118), in the following analysis, we assume that it is some increasing sequence and we will specify $B_n$ later.

$$\hat{F}_{\text{trunc},n}(x) := \frac{1}{n} \sum_{i=1}^{n} \mathbb{I}[\text{sign}(X_i) \min(|X_i|, B_n^{\frac{1}{p}}) < x] \tag{120}$$

$$F_{\text{trunc},n}(x) := \mathbb{P}[\text{sign}(X_i) \min(|X_i|, B_n^{\frac{1}{p}}) < x]. \tag{121}$$

Note that $F_{\text{trunc},n}$ is discountinuous at $x = -B_n$ and $x = B_n$. Hence, we have,

$$F_{\text{trunc},n}(x) = \begin{cases} 1 & x \geq B_n, \\ F(x) & x \in [-B_n, B_n), \\ 0 & x < -B_n. \end{cases} \tag{122}$$

We first split the analysis into two parts,

$$W_1(\hat{F}_{\text{trunc},n}, F_{\text{trunc},n}) = \int_{-\infty}^{\infty} |\hat{F}_{\text{trunc},n}(x) - F_{\text{trunc},n}(x)| dx \tag{123}$$

$$= \int_{-B_n}^{B_n} |\hat{F}_{\text{trunc},n}(x) - F_{\text{trunc},n}(x)| dx \tag{124}$$

$$\leq 2B_n \cdot \sup_{x \in [-B_n, B_n]} |\hat{F}_{\text{trunc},n}(x) - F_{\text{trunc},n}(x)| \tag{125}$$

$$\leq 2B_n \sqrt{\frac{\ln(C/\delta)}{2n}} \quad (\because \text{ DKW inequality}) \tag{126}$$

$$W_1(F_{\text{trunc},n}, F) = \int_{-\infty}^{\infty} |F_{\text{trunc},n}(x) - F(x)| dx \tag{127}$$

$$= \int_{-\infty}^{-B_n} |F_{\text{trunc},n}(x) - F(x)| dx + \int_{-B_n}^{B_n} |F_{\text{trunc},n}(x) - F(x)| dx \tag{128}$$

$$+ \int_{B_n}^{\infty} |F_{\text{trunc},n}(x) - F(x)| dx \tag{129}$$

$$\leq \int_{-\infty}^{-B_n} F(x) dx + \int_{-B_n}^{B_n} |F_{\text{trunc},n}(x) - F(x)| dx + \int_{B_n}^{\infty} 1 - F(x) dx \tag{130}$$

$$\leq \int_{B_n}^{\infty} F(-x)dx + \int_{B_n}^{\infty} 1 - F(x)dx \tag{131}$$

$$\leq 2 \int_{B_n}^{\infty} \mathbb{P}\left(|X| > x\right) dx \leq 2 \int_{B_n}^{\infty} \frac{\nu_p}{x^p} dx \tag{132}$$

$$= -\frac{2\nu_p}{(p-1)} x^{-(p-1)} \Big|_{B_n}^{\infty} = \frac{2\nu_p}{(p-1)B_n^{p-1}} \tag{133}$$

Combining these, the total deviation is

$$W_1(\hat{F}_{\text{trunc},n}, F) \leq W_1(\hat{F}_{\text{trunc},n}, F_{\text{trunc},n}) + W_1(F_{\text{trunc},n}, F) \tag{134}$$

$$\leq 2B_n \sqrt{\frac{\ln(C/\delta)}{2n}} + \frac{2\nu_p}{(p-1)B_n^{p-1}} \tag{135}$$

The optimal truncation level $B_n$ minimizes the combined deviation terms. Solving for $B_n$ gives

$$B_n := \left(\frac{\nu_p}{p-1}\right)^{\frac{1}{p}} \cdot \left(\frac{2n}{\ln(C/\delta)}\right)^{\frac{1}{2p}} \tag{136}$$

Then, the confidence interval of deviation inequalities is computed as

$$W_1(\hat{F}_{\text{trunc},n}, F) \leq 4 \left(\frac{\nu_p}{p-1}\right)^{\frac{1}{p}} \cdot \left(\frac{\ln(C/\delta)}{2n}\right)^{\frac{1}{2}\left(1-\frac{1}{p}\right)} \tag{137}$$

For a Lipschitz risk measure $\rho$ with Lipschitz constant $\kappa$, we can derive the confidence interval for $\rho$ as

$$\rho(\hat{F}_{\text{trunc},n}) - \rho(F) \leq 4\kappa \left(\frac{\nu_p}{p-1}\right)^{\frac{1}{p}} \cdot \left(\frac{\ln(C/\delta)}{2n}\right)^{\frac{1}{2}\left(1-\frac{1}{p}\right)} \tag{138}$$

This result provides a robust confidence interval for risk measure aware bandit algorithms under heavy-tailed settings. However, the derived order of convergence, $\frac{1}{2}(1 - \frac{1}{p})$, is worse than the ideal order of $1 - \frac{1}{p}$. This highlights a limitation of the truncated empirical distribution approach, as the truncation mitigates extreme values but sacrifices optimal convergence rates for risk measure estimation.

