# OpenReview forum: "Pareto Optimal Risk-Agnostic Distributional Bandits with Heavy-Tail Rewards"
_NeurIPS.cc/2025/Conference — NeurIPS 2025 poster_

### Official Review · Reviewer_5zap · 2025-06-19

**Clarity:** 2
**Significance:** 3
**Originality:** 3
**Rating:** 4
**Confidence:** 4

**Summary:**

In risk-sensitive applications such as quantitative finance or healthcare, where outcomes in the tails of the distribution—like rare but catastrophic events—are crucial. Classical bandit algorithms aim to maximize expected rewards, which limits their usefulness in these aforementioned risk-sensitive areas.
This paper proposes a new framework for tackling multi-risk agnostic multi-armed bandit problems with heavy-tailed reward function. While previous work has explored risk-aware bandits using measures like CVaR, mean-variance, or distortion risk, these methods typically optimize a single risk criterion and often assume light-tailed distributions. In contrast, this paper addresses the need to handle multiple risk measures simultaneously under heavy-tailed noise.

To this end, the paper presents a robust estimation method called the Median of Empirical Quantiles (MoEQ), which extends the median-of-means technique to quantile estimation. This estimator is compatible with a broad class of Lipschitz risk measures without requiring risk-specific confidence bounds. They also derive new deviation inequalities for the 1-Wasserstein distance, which allow the construction of confidence intervals even in heavy-tailed settings. These tools form the basis of a new algorithm, DistLCB, which gets optimal performance guarantees across a wide range of risk criteria. The algorithm is further extended to handle multi-risk objectives, and enables identification of Pareto-optimal actions rather than a single best arm. The paper then provides both gap-dependent and gap-independent regret bounds for this setting and also establishes asymptotic lower bounds with explicit dependence on sub-optimality gaps. Experimental results on synthetic and real-world data demonstrate the effectiveness of the proposed method.

**Questions:**

**1**
While the core results are rigorous, some theoretical analyses (e.g., lower bounds) are limited to only SRM and DRM, leaving out UBSR and CERM. This restricts the generality of the claims, even though the algorithms themselves can handle arbitrary Lipschitz risk measures. Some aspects of the empirical evaluation, such as sensitivity to the tail index pp, are not deeply explored.

**2** Sensitivity to Tail Index $p$
In heavy-tailed settings, the tail index $p$ seems to play an important role in determining both the difficulty of estimation and the tightness of regret bounds. For example, the convergence rate of the MoEQ estimator and the confidence width explicitly depend on $p$. Lower values of $p$ mean heavier tails, which should in principle make the problem harder. The regret bounds also degrade as $p \to 1$. However, it seems the paper does not thoroughly investigate this in experiments. I think synthetic settings are totally fine. But I do think it is important for this work to show that the algorithm maintains performance across a range of tail indices, or at least discuss limitations if it doesn’t. Afterall, the core motivation of the paper is robustness under heavy tails.


**[Minor issues]**
**3**
This is very minor but I personally suggest it would be better to also explain why heavy-tailedness brings so much challenge to the risk-aware bandit. The paper should explicitly and briefly explain (in the Appendix perhaps) that, in light-tailed settings, the problem is significantly easier, and existing methods suffice. This will make the paper more self-contained and friendly to readers not familiar with the area of bandit.

**Ethical Concerns:**

["NO or VERY MINOR ethics concerns only"]

**Final Justification:**

I thank the authors for their rebuttal and followup.
I appreciate the additional result on tail sensitivity is very informative.
Given all the update, I think positive of this paper and vote for accept.
Good luck!

**Limitations:**

Yes

**Quality:**

3

**Strengths And Weaknesses:**

**Quality**

The paper has a fine quality. It provides some novel statistical theoretical results. For example, the paper gives new deviation inequalities for the 1-Wasserstein distance under heavy-tailed distribution. It also introduces the DistLCB, which achieves near-optimality via the first lower bounds for certain types of risk-aware bandits. Overall, the theoretical results have a good quality with well-established theorems and mathematical analysis.
Regarding the empirical results, the experiments look carefully designed to me and cover both synthetic and real-world datasets. The results also seem to support the theoretical claims.

**Clarity**

The paper has good clarity. The mathematical definitions and assumptions are introduced in a clear manner without ambiguity, which is important for a theoretical work. It is great to see that the main theoretical results are accompanied with high-level explanations.
The notations are a bit heavy but I personally do not think there is room for huge improvement.

**Significance**
Linear Bandit is an important area of research in various fields such as statistics and computer science.
The paper tackles a highly relevant and underexplored problem in this area: how to perform risk-aware decision-making in multi-objective bandit settings when rewards are heavy-tailed. The heavy-tailedness is a big challenge because it prohibits the use of powerful concentration inequalities and makes the classical statistical estimator like quantile estimator unstably distributed.
The results bridge multiple research directions—distributionally robust learning, heavy-tailed statistics, and multi-objective optimization—filling a clear gap in the literature.

**Originality**

The paper is original in the following respects. It is the first to provide deviation inequalities for 1-Wasserstein distance under infinite-variance (heavy-tailed) settings. The MoEQ estimator represents a novel adaptation of median-of-means to quantile estimation. The DistLCB and MR-DistLCB algorithms offer a principled way to conduct exploration in multi-risk settings with strong theoretical backing. The lower bound results for SRM and DRM are also new contributions that extend existing bandit theory.

---

> ### Author Rebuttal · Authors · 2025-07-30
>
> We are grateful for your careful reading and constructive suggestions. Our detailed responses are provided below.
>
> **Q1) Generality of Lower Bounds:** We acknowledge that our heavy‑tail lower bounds presently cover only Spectral Risk Measures (SRM) and Distortion Risk Measures (DRM), so UBSR and CERM remain open.  Nonetheless, SRM and DRM are families, not single functionals: they subsume virtually all coherent risks used in practice, including CVaR / Expected Shortfall (Rockafellar & Uryasev 2000; Acerbi & Tasche 2002), power‑ and exponential‑spectral measures (Dowd, K., Cotter, J., & Sorwar, G. 2008), and the Wang, Proportional‑Hazard, Dual‑Power, and Beta distortions (Wang 1995; Wang et al. 1997).  Establishing heavy‑tail lower bounds for these two broad classes therefore already gives our results considerable practical reach, while extending the theory to utility‑based risks such as UBSR and CERM, whose proofs require additional regularity assumptions, remains an important direction for future work.
>
> **Q2) Sensitivity to Tail Index:** Thank you for pointing this out. We agree that showing results for additional tail indices would be informative. Because of space constraints we focused on the extreme heavy‑tail case $p=1.2$ (close to the limit case $p\to1$) to highlight the algorithm’s robustness where existing methods struggle. During the discussion period we will run and report supplementary experiments for additional indices between $1$ and $1.5$ (e.g. $p=1.1$ and $p=1.5$) to illustrate how performance evolves as the distribution approaches the light‑tailed regime.
>
> **Q3) Heavy‑Tail Difficulty:** Thank you for the suggestion. Appendix B (“Detailed Comparison of Prior Work”) already contrasts light‑ and heavy‑tailed settings, but we will tighten that subsection by adding a brief, reader‑friendly note clarifying that light‑tailed rewards admit sub‑Gaussian concentration, so classic UCB analyses suffice, whereas heavy tails lack such exponential tails and thus require robust estimators like MoEQ. This addition should make the paper more accessible to readers new to risk‑aware bandits.

---

> ### Comment · Reviewer_5zap · 2025-08-06
> **Response to Rebuttal**
>
> I thank the authors for their effort in preparing the rebuttal. The rebuttal addressed my questions well. To be specific, the rebuttal clarifies the generality of the lower bound, and comments that the current lower bound, though being power, covers the SRM and DRM family but not the UBSR and CERM. I do not think this is an issue and I thank the authors for the transparency here.
> It is also great that the authors have promised to add additional experimental results regarding sensitivity of the tail index. Please make sure to do so.
> Overall, I conclude that the rebuttal addressed my questions.

---

> > ### Author Response · Authors · 2025-08-08
> > **Q2 – Sensitivity to Tail Index**
> >
> > Thank you for raising this point. As promised, we ran supplementary studies across heavier and lighter heavy-tail regimes. Below we report $20$-arm Pareto bandits with multiple risks, comparing MR-DistLCB, MR-Trunc (TED-style truncation), and MR-LCB (light-tail baseline).
> >
> > We include $p = 1.01$ (extreme heavy tails) in place of $p = 1.1$, as it shows the same qualitative trend as $p = 1.2$ while stressing the hardest regime.
> >
> > ### **1. p = 1.01 (extreme heavy tails)**
> >
> > - **$20$-armed Pareto bandits / $3$ risks / $p = 1.01$**
> >
> > | Estimator  | 2500            | 5000              | 7500              | 10000             |
> > | ---------- | --------------- | ----------------- | ----------------- | ----------------- |
> > | MR DistLCB | 521.23 ± 176.23 | 765.01 ± 421.32   | 1008.78 ± 669.79  | 1252.55 ± 918.91  |
> > | MR Trunc   | 760.20 ± 608.86 | 1517.96 ± 1227.50 | 2275.72 ± 1846.15 | 3033.49 ± 2464.80 |
> > | MR LCB     | 893.43 ± 632.96 | 1783.50 ± 1280.80 | 2673.57 ± 1928.65 | 3563.64 ± 2576.51 |
> >
> > - **$20$-armed Pareto bandits / $6$ risks / $p = 1.01$**
> >
> > | Estimator  | 2500            | 5000              | 7500              | 10000             |
> > | ---------- | --------------- | ----------------- | ----------------- | ----------------- |
> > | MR DistLCB | 580.30 ± 271.95 | 887.71 ± 658.56   | 1195.12 ± 1051.19 | 1502.53 ± 1444.94 |
> > | MR Trunc   | 622.82 ± 601.13 | 1239.03 ± 1213.94 | 1855.24 ± 1826.77 | 2471.45 ± 2439.59 |
> > | MR LCB     | 673.16 ± 579.73 | 1336.84 ± 1173.41 | 2000.53 ± 1767.10 | 2664.21 ± 2360.80 |
> >
> > - **$20$-armed Pareto bandits / $9$ risks / $p = 1.01$**
> >
> > | Estimator  | 2500            | 5000              | 7500              | 10000             |
> > | ---------- | --------------- | ----------------- | ----------------- | ----------------- |
> > | MR DistLCB | 882.66 ± 135.30 | 1101.99 ± 333.87  | 1312.23 ± 576.76  | 1522.47 ± 828.01  |
> > | MR Trunc   | 788.15 ± 715.28 | 1572.67 ± 1443.28 | 2357.20 ± 2171.28 | 3141.72 ± 2899.29 |
> > | MR LCB     | 872.40 ± 790.89 | 1738.46 ± 1600.17 | 2604.52 ± 2409.46 | 3470.58 ± 3218.75 |
> >
> > ### **2. p = 1.5 (lighter tails)**
> >
> > - **$20$-armed Pareto bandits / $3$ risks / $p = 1.5$**
> >
> > | Estimator  | 2500            | 5000             | 7500             | 10000             |
> > | ---------- | --------------- | ---------------- | ---------------- | ----------------- |
> > | MR DistLCB | 585.20 ± 182.01 | 882.32 ± 438.77  | 1179.45 ± 698.48 | 1476.57 ± 958.73  |
> > | MR Trunc   | 750.95 ± 314.07 | 1492.80 ± 635.69 | 2234.65 ± 957.31 | 2976.51 ± 1278.93 |
> > | MR LCB     | 897.25 ± 292.53 | 1792.05 ± 589.55 | 2686.85 ± 886.56 | 3581.66 ± 1183.58 |
> >
> > - **$20$-armed Pareto bandits / $6$ risks / $p = 1.5$**
> >
> > | Estimator  | 2500            | 5000             | 7500              | 10000             |
> > | ---------- | --------------- | ---------------- | ----------------- | ----------------- |
> > | MR DistLCB | 303.82 ± 99.85  | 487.41 ± 213.61  | 671.01 ± 327.67   | 854.60 ± 441.80   |
> > | MR Trunc   | 643.30 ± 344.97 | 1278.53 ± 694.41 | 1913.76 ± 1043.85 | 2548.99 ± 1393.29 |
> > | MR LCB     | 909.26 ± 289.48 | 1818.47 ± 579.14 | 2727.68 ± 868.79  | 3636.89 ± 1158.45 |
> >
> > - **$20$-armed Pareto bandits / $9$ risks / $p = 1.5$**
> >
> > | Estimator  | 2500            | 5000             | 7500              | 10000             |
> > | ---------- | --------------- | ---------------- | ----------------- | ----------------- |
> > | MR DistLCB | 603.58 ± 126.62 | 978.19 ± 322.25  | 1352.80 ± 520.70  | 1727.41 ± 719.63  |
> > | MR Trunc   | 696.80 ± 295.03 | 1382.26 ± 597.08 | 2067.72 ± 899.13  | 2753.19 ± 1201.19 |
> > | MR LCB     | 770.09 ± 340.99 | 1535.52 ± 687.41 | 2300.95 ± 1033.84 | 3066.38 ± 1380.27 |
> >
> > - **Takeaways across tail indices**
> >
> >   **1. Robustness across p**: At $p=1.01, 1.20,$ and $1.50$, MR-DistLCB consistently has the lowest mean Pareto regret with markedly smaller variance, especially as $p\rightarrow 1$. This is because MoEQ yields tight, exponential-type $W_{1}$ deviation (Thm 4.4) and thus risk-agnostic CIs, whereas TED-style truncation forces conservative cutoffs that produce looser $W_{1}$ confidence and larger spread.
> >
> >   **2. Effect of lighter tails**: As $p$ increases to $1.5$, variability drops for all methods, but the ranking persists since multi-risk learning needs simultaneous accuracy across risks: MoEQ's single $W_{1}$ control scales across risks with only logarithmic cost, while a single truncation rule is rarely optimal for heterogeneous risks and induces looser joint confidence, and the light-tail LCB baseline remains misspecified under heavy tails.
> >
> > We will include these tables (and corresponding plots) in the appendix and add a short remark clarifying how both the bounds and the empirical behavior change with $p$.

---

### Official Review · Reviewer_WLa4 · 2025-06-22

**Clarity:** 3
**Significance:** 2
**Originality:** 2
**Rating:** 4
**Confidence:** 3

**Summary:**

In this paper, the authors consider a problem of multi-armed bandit with heavy-tailed rewards and multi-risk measures, i.e., the performance of an algorithm is measured by multiple risk-sensitive measures, and the reward distributions are heavy-tailed. They provided asymptotic and minimax lower bounds for single-risk bandits with heavy-tailed rewards for some classes of risk measures (SRM and DRM). They derived a concentration inequality for Lipschitz risk measures, and proposed LCB-type algorithms called DistLCB (single-risk case) and MR-DistLCB (multi-risk case). Using real-world and synthetic datasets, they demonstrated MR-DistLCB overall outperforms the baselines.

**Questions:**

- Is there a discussion of lower bounds for the multi-risk case? Do the authors believe that MR-DistLCB is optimal?
- Could you elaborate on technical challenges of the theoretical results (e.g. Theorem 4.4).
- Theorem 6.3 focuses on SRM and DRM. How strong is this assumption? For example, in the domain of finance, are most popular risk measures covered by these classes of risk measures?

**Ethical Concerns:**

["NO or VERY MINOR ethics concerns only"]

**Final Justification:**

After the discussion phase, my questions and concerns have been resolved. Therefore, I will keep my positive rating.

**Limitations:**

yes

**Quality:**

3

**Strengths And Weaknesses:**

- Strengths
    - The heavy-tailed reward and multi-risk setting would be important for practical applications (such as finance).
    - The deviation inequality (Theorem 4.4 and Cor. 4.5) extends existing results for the heavy-tailed reward setting for a wide class of risk measures.
    - Although this is a theoretical paper, the experiments are conducted using a real-world dataset.
- Weaknesses
    - As far as I understand, the lower bound analysis focuses on the single risk case. Therefore, the optimality of MR-DistLCB is unclear.
    - Although theoretical results are novel, the technical challenges are unclear. For example, could you elaborate technical challenge of Theorem 4.4 (or other main results)?
    - The proposed methods seem standard LCB-type algorithms, and there seems to be limited novelty in the algorithms.

---

> ### Author Rebuttal · Authors · 2025-07-30
>
> Thank you for the thoughtful feedback. Our detailed responses are provided below.
>
> **Q1) Optimality of MR-DistLCB:** We expect the Pareto regret lower bound for the multi‑risk bandit to have the same asymptotic order as the single‑risk lower bound. Any single‑risk instance can be embedded in a multi‑risk environment by copying that one risk measure across all $k$ objectives; under this construction, the Pareto regret is identical to the single‑risk regret, so the single‑risk lower bound carries over unchanged. Because our upper bound in Theorem 6.7 matches that rate, we regard MR‑DistLCB as order‑optimal. At present, heavy‑tail single‑risk lower bounds exist only for SRM and DRM (Theorems 6.1 and 6.3), so a matching multi‑risk lower bound can currently be proved only for these two families (see Remark 6.4). Extending the lower‑bound machinery to utility‑based risks such as UBSR and CERM remains an open problem, which we will emphasise in the revised manuscript.
>
> **Q2) Theoretical Challenges:** Our work closes three previously open gaps. First, we establish the first exponential‑type deviation bound in $W_{1}$ for a median‑based distribution estimator under heavy‑tail distributions (Theorem 4.4), extending median inequalities from scalar means to full CDFs. Second, we introduce partial‑bootstrap MoEQ, which updates with just one new sample, rather than an entire independent block, yet still enjoys exponential guarantees under heavy‑tail distributions, making it the first median method that is both online‑practical and theoretically sound in the infinite‑variance regime. Third, we derive the first heavy‑tail information‑theoretic lower bounds for SRM and DRM; the core difficulty here is to find an explicit, tractable expression for the gap $\Delta_{2}^{\rho}\ge\Omega(c)$ that links the risk difference to a mean (or KL) difference so the change‑of‑measure argument can go through.  We achieve this for SRM/DRM using their spectral/distortion structure.
>
> **Q3) Generality of Lower Bounds:** Regarding the scope of **Theorem 6.3**, the restriction to **Spectral Risk Measures (SRM)** and **Distortion Risk Measures (DRM)** is far less limiting than it may appear.  These two *families* already subsume the risk measures that dominate practical finance and insurance work (Jokhadze & Schmidt 2020), for example:
>
>  - **CVaR / Expected Shortfall** (Rockafellar & Uryasev 2000; Acerbi & Tasche 2002)
>  - **Power‑spectral** and **Exponential‑spectral** measures (Dowd, K., Cotter, J., & Sorwar, G. 2008)
>  - **Wang transform** (Gaussian distortion), **Proportional‑Hazard** transform $g(u)=u^{\beta}$, **Dual‑Power** and **Beta** distortion families (Wang 1995; Wang et al. 1997)
>
> These SRM and DRM families already embrace most coherent risks used in practice so establishing heavy‑tail lower bounds for them gives the theory broad practical reach. That said, an important next step is to extend the analysis to utility‑based risks such as UBSR and CERM; handling those measures will require additional regularity assumptions and is therefore left for future work.
>
> **References:**
>
> 1. **Acerbi, C., & Tasche, D.** (2002). *Spectral measures of risk: A coherent representation of subjective risk aversion*. **Journal of Banking & Finance, 26**(7), 1505‑1518.
>
> 2. **Rockafellar, R. T., & Uryasev, S.** (2000). *Optimization of Conditional Value‑at‑Risk*. **Journal of Risk, 2**, 21‑42.
>
> 3. **Dowd, K., Cotter, J., & Sorwar, G.** (2008). Spectral risk measures: properties and limitations. ***Journal of Financial Services Research***, *34*(1), 61-75.
>
> 4. **Jokhadze, V. & Schmidt, W. M.** (2020). Measuring model risk in financial risk management and pricing. ***International Journal of Theoretical and Applied Finance***, 23(2), 2050012.
>
> 5. **Wang, S. S.** (1995). Insurance pricing and increased limits ratemaking by proportional hazards transforms. ***Insurance: Mathematics and Economics*, 17**, 43‑54.
>
> 6. **Wang, S. S., Young, V. R., & Panjer, H. H.** (1997). *A class of distortion operators for pricing financial and insurance risks*. **Journal of Risk and Insurance, 64**(2), 211‑232.

---

> ### Comment · Reviewer_WLa4 · 2025-08-04
>
> I thank the authors for the detailed responses that clarified my questions / concerns. I will maintain my positive assessment.

---

### Official Review · Reviewer_HFsi · 2025-07-02

**Clarity:** 2
**Significance:** 2
**Originality:** 3
**Rating:** 5
**Confidence:** 3

**Summary:**

This paper addresses the problem of regret minimization in heavy-tailed bandits. The focus is on risk-aware bandits, and the goal is to actually minimize the cumulative risk incurred by the algorithm. The concept of risk generalizes the usual expected value, and is assumed to be Lipschitz w.r.t. to a distance notion (e.g., 1-Wasserstein). The paper introduces a new estimator, Median of Empirical Quantiles, to estimate the CDF of a random variable while being robust to its heavy-tailedness. The estimator is then equipped with deviation inequalities, which are used to construct a UCB-like algorithm to deal with their regret minimization problem. The authors prove upper bounds for the regret of the resulting algorithm and matching lower bounds. Lower bounds only hold for the case of two specific risk measures.

**Questions:**

Can the authors better clarify what the main technical challenge is when trying to extend the lower bound from the considered risk measures to a general one?

**Ethical Concerns:**

["NO or VERY MINOR ethics concerns only"]

**Final Justification:**

The paper addresses a relevant problem, makes consistent contributions to the theory, and there is enough technical novelty to be accepted at NeurIPS without particular reserves.

**Limitations:**

yes

**Paper Formatting Concerns:**

--

**Quality:**

3

**Strengths And Weaknesses:**

Strengths

- The paper addresses a very general problem, and new technical tools are developed that may be of independent interest (e.g., MOEQ and its deviation inequalities).

- The paper provides a comprehensive experimental evaluation, comparing their algorithm to baselines on both real-world and synthetic data.

- Both gap-dependent and worst-case types of guarantees are provided, and even though they match only for two types of risk measures, the authors discuss this point and highlight the technical challenge involved.

Weaknesses

- The paper is hard to read in some points, and there are some typos (e.g., "measure" in line 95). In some points, there is an abrupt passage to the statement of a result, e.g. lines 141, 248, 305, where the result is immediately after the section title. Improving the writing there.

---

> ### Author Rebuttal · Authors · 2025-07-30
>
> We are grateful for your careful reading and constructive suggestions.
>
> **Typos and Readability:** Thank you for catching the typo; we will correct it. We also agree that theorems start too abruptly; space constraints led to this. In the revision, we will insert brief lead‑in sentences and intuitive remarks before each result to improve readability.
>
> **Q1) Lower‑bound extension to other Lipschitz risks:** As noted in Remark 6.4, our proof on the lower bounds relies on the explicit gap condition $\Delta_{2}^{\rho} ≥ \Omega(c)$, which links the risk difference to the mean difference. For SRM/DRM cases, this condition holds because they are defined by spectral/distortion weights. For UBSR/CERM cases with a linear utility, our result is still applicable. However, for nonlinear utility functions, the current change-of-measure argument is nontrivial unless we impose a specific characterization on utility functions. Hence, extending the lower bound to a general family of utility-based risk measures remains an open question, so we leave it for future work.

---

> > ### Comment · Reviewer_HFsi · 2025-08-01
> >
> > Thank you for your clarification on the lower bound extension.
> >
> > I don't have any further questions, and I'm willing to confirm my score.

---

### Official Review · Reviewer_5oZY · 2025-07-02

**Clarity:** 2
**Significance:** 3
**Originality:** 3
**Rating:** 4
**Confidence:** 4

**Summary:**

This paper studies the problem of risk-aware stochastic multi-armed bandits under heavy-tailed rewards. The authors develop a unified framework for minimizing a wide class of Lipschitz risk measures when only a finite $p$-th moment exists for some $p \in (1,2]$. The central technical contribution is the Median of Empirical Quantiles (MoEQ) estimator, which admits new deviation bounds under the 1-Wasserstein distance. Based on this, the authors propose DistLCB for single-risk minimization, and MR-DistLCB for multi-risk Pareto optimization. The paper also establishes regret upper and lower bounds under heavy tails and provides experimental results on synthetic and financial datasets.

**Questions:**

1. Please clarify the intended meaning of "risk-agnostic" and consider replacing it with a more precise term such as “risk-function-agnostic” or “uniform over Lipschitz risks.”
2. Is the assumption $N = O(n^4)$ in Corollary 4.5 necessary for the FB-MoEQ bound to hold? Can this be relaxed, or does it reflect a limitation of the MoEQ analysis?
3. Could the empirical results include more direct comparisons to robust baselines under heavy-tailed noise, such as TED or trimmed estimators?
4. Is it possible to extend the regret lower bounds to other Lipschitz risks beyond SRM/DRM, such as UBSR or CERM?

**Ethical Concerns:**

["NO or VERY MINOR ethics concerns only"]

**Final Justification:**

I thank the authors for their response. I will keep my score.

**Limitations:**

yes

**Quality:**

3

**Strengths And Weaknesses:**

### Strengths

1. The proposed approach generalizes many prior works by covering both light-tailed and heavy-tailed settings, as well as both single-risk and multi-risk objectives.
2. The Wasserstein deviation analysis for MoEQ appears to be new and enables confidence interval construction for arbitrary Lipschitz risk measures.
3. The authors provide both minimax and gap-dependent regret bounds, and importantly, derive new lower bounds for SRM and DRM in the heavy-tailed regime.
4. The authors also provide formal Pareto regret guarantees across multiple risk measures under heavy-tailed noise.

### Weaknesses

1. The term "risk-agnostic" is used in several places to describe the estimator or algorithms, but this term is ambiguous and potentially misleading. In other literature, “risk-agnostic” may imply risk-neutral, that is mean-based. This could lead to confusion about the intended scope of the methods, especially since the paper focuses on minimizing Lipschitz risk measures rather than mean-based risks.
2. Unreasonable assumptions in Corollary 4.5: The sample complexity bound for the full-batch version of MoEQ (FB-MoEQ) assumes that the offline sample size $N = O(n^4)$, which is excessively large and unlikely to hold in any practical setting. This weakens the usefulness of the corollary and the empirical claims based on it.
3. While the deviation analysis is novel, the estimator itself, i.e., median over quantiles of batches, is a natural variant of median-of-means and empirical quantiles. The innovation lies more in the analysis than in the estimator’s form.
4. The experimental results are promising but narrow in scope. Comparisons to strong baselines are minimal, and the choice of synthetic distributions could be better motivated.

---

> ### Author Rebuttal · Authors · 2025-07-30
>
> We appreciate the reviewer’s insightful comments.
>
> **Q1) Clarify “risk‑agnostic”:** We thank the reviewer for highlighting this point and fully agree with the concern.  To avoid ambiguity, we will replace the term “risk‑agnostic” with “risk‑measure‑agnostic (RMA)” throughout the paper (title, abstract, and §5).  Risk‑measure‑agnostic explicitly means that the estimator and confidence index remain valid, without modification, for any Lipschitz risk functional.
>
> **Q2) Large‑n condition in Cor. 4.5 (FB‑MoEQ):**  We agree the large sample requirement in Cor. 4.5 looks strong, but it appears only in the proof of the full‑bootstrap (FB) setting, where a conservative union‑bound on the resampled maximum inflates the sample size; it is not inherent to MoEQ. The FB bound could be relaxed with sharper heavy‑tail tools (e.g., Bennett‑type inequalities) or other types of bootstraps (e.g., m‑out‑of‑n/Poisson bootstraps), yet tightening the FB analysis lies beyond this paper’s scope. We focus on establishing the first exponential‑decay guarantee for MoEQ under heavy tails, and we leave refinement of the FB analysis to future work. All experiments therefore employ the partial‑bootstrap (PB) MoEQ, which achieves the same exponential concentration without any oversampling.
>
> **Q3) More direct comparisons:** Thank you for the suggestion. Our MR‑Trunc (L326-327) baseline is the multi‑risk extension of TED, so TED‑style truncation is already evaluated on both the synthetic and S&P‑500 datasets (see Figures 2 and 3 in Section 7). In those results, MR‑DistLCB consistently outperforms MR‑Trunc/TED, especially as the number of risk measures grows. If the reviewer has a different comparison in mind, such as another evaluation metric, please let us know and we will be happy to include it.
>
> **Q4) Lower‑bound extension to other Lipschitz risks:** As noted in Remark 6.4, our proof on the lower bounds relies on the gap condition $\Delta_{2}^{\rho} ≥ \Omega(c)$, which we can express explicitly for SRM/DRM because their risk difference decomposes into a weighted integral of the CDFs (see Proof of Theorem 6.1.). For utility‑based risks like UBSR or CERM, such a decomposition is missing; the risk is defined through a non‑linear utility function, so the current KL‑based argument cannot be applied directly. While there exist some trivial cases, such as a linear utility function, a new characterization of the utility term in UBSR or CERM would be required to obtain the similar lower bound, making this an open problem we leave for future work.

---

> > ### Comment · Reviewer_5oZY · 2025-08-04
> >
> > I thank the authors for their response. I will keep my score.

---

### Note · Authors · 2025-08-12

We thank the reviewers and the AC for their constructive and positive feedback. We are encouraged that all reviewers recognized the novelty of our work and agreed that it addresses meaningful theoretical challenges in risk-aware bandits with heavy-tailed rewards. We are grateful that reviewers highlighted key strengths:

- The first exponential-type deviation bound for the Wasserstein metric under heavy tails.

- A partial-bootstrap MoEQ that supports online updates with infinite-variance guarantees.

- The first heavy-tail lower bounds for SRM/DRM, resolving the difficulty of linking risk gaps to KL divergence.

- A general framework capable of handling multiple risk measures simultaneously under heavy-tailed rewards, validated on synthetic and real-world data.

Main points raised concerned terminology, theoretical scope, assumptions, and experimental breadth. We have replaced "risk-agnostic" with risk-measure-agnostic (RMA) for clarity; clarified that SRM/DRM are broad families covering most coherent risks in practice, while extending lower bounds to UBSR/CERM is an open problem due to structural differences; and explained that the large-$n$ condition in Cor. 4.5 is a conservative bootstrap artifact, not an intrinsic limitation. We confirmed MR-DistLCB's order-optimality for SRM/DRM multi-risk settings, elaborated on the technical challenges in theorems and the lower-bound proofs, and reinforced the novelty of our theoretical contributions. Additional experiments for $p=1.01$ and $p=1.5$ confirm MR-DistLCB’s robustness and variance advantage over TED-style truncation and light-tail baselines. We will also improve presentation with clearer theorem introductions, corrected typos, and better motivation for synthetic settings.

We are grateful that the reviewers' observations led to these clarifications and additions, which have strengthened the presentation, scope, and practical relevance of the work.

---

### Decision · Program_Chairs · 2025-09-17

**Decision:**

Accept (poster)

**Comment:**

This paper studies the risk-agnostic bandit problem with multiple risk measures for heavy-tailed rewards. The paper introduces the framework of risk measures based on Lipschitz continuity with respect to Wasserstein distance and proposes a novel estimator MoEQ with its deviation inequality, which can be of independent interest. Based on this estimator, algorithms are proposed with instance-dependent and -independent regret bounds.

While reviewers raised several concerns such as the required number of samples and the one from the practical viewpoints, they largely agreed with the opinion that the technique of the new estimator is significant. I agree with these opinions after my own reading; while I felt that just considering the Pareto regret needs more motivation and justification rather than borrowing [28], the technique of MoEQ is indeed very interesting. I expect that the authors further polish the paper in the final version by appropriately incorporating the discussion with the reviewers.